# HandMeThat: Human-Robot Communication in Physical and Social Environments

**Yanming Wan**[*]
IIIS, Tsinghua University

**Jiayuan Mao**[*]
MIT CSAIL

**Joshua B. Tenenbaum**
MIT BCS, CBMM, CSAIL

## Abstract

We introduce HandMeThat, a benchmark for a holistic evaluation of instruction understanding and following in physical and social environments. While previous datasets primarily focused on language grounding and planning, HandMeThat considers the resolution of human instructions with *ambiguities* based on the physical (object states and relations) and social (human actions and goals) information. HandMeThat contains 10,000 episodes of human-robot interactions. In each episode, the robot first observes a trajectory of human actions towards her internal goal. Next, the robot receives a human instruction and should take actions to accomplish the subgoal set through the instruction. In this paper, we present a textual interface for our benchmark, where the robot interacts with a virtual environment through textual commands. We evaluate several baseline models on HandMeThat, and show that both offline and online reinforcement learning algorithms perform poorly on HandMeThat, suggesting significant room for future work on physical and social human-robot communications and interactions.

## 1 Introduction

To collaborate with human partners successfully in complex environments, robots should be able to interpret and follow natural language instructions in contexts. Consider the example shown in Fig. 1, a human is preparing fruits for bottling. In the middle of her actions, the human asks a robot agent for help: "can you hand me that one on the table please?" The robot needs to correctly interpret the sentence in the current context and interact with objects to accomplish this task.

Here, the human utterance essentially specifies a subgoal for the robot (getting the knife), derived from her own goal (bottling fruits). In reality, such subgoal can be under-specified in human utterances, typically for two reasons. First, the human assumes that the robot has knowledge about her goal [1, 2]. Second, human makes trade-offs between accuracy and efficiency of communication [3, 4, 5]. While previous benchmarks in similar domains have been primarily focusing on the language grounding of object properties (e.g., "table"), relations (e.g., "on"), and planning (e.g., object search and manipulation) [6, 7], in this paper, we highlights the additional challenge for understanding human instructions with *ambiguities* (i.e., recognizing the subgoal) based on physical states and human actions and goals. In this example, the human has just taken fruits out of the refrigerator and is trying to slice them on the countertop. Thus, the object to be retrieved should be the knife.

Developing a benchmark for resolving ambiguous instructions based on both physical and social environments is challenging in both data collection and automatic evaluation: it is hard to collect the everyday dialogues with corresponding physical states, and it is hard to build automatic evaluation protocols that involve human judgments about robot success. In this paper, we present a new benchmark, HandMeThat, aiming for a holistic evaluation of language instruction understanding and

---

[*]indicates equal contribution. Correspondence to: `jiayuanm@mit.edu`.
Project website: http://handmethat.csail.mit.edu/

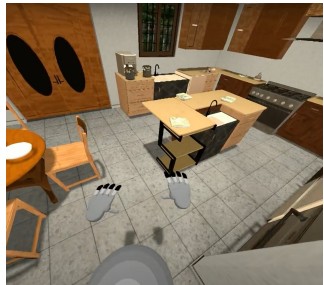 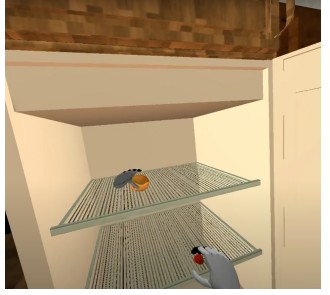 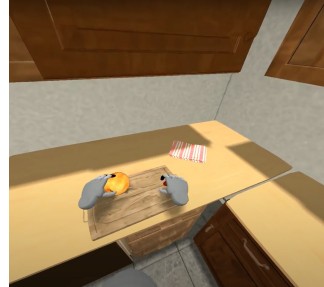

1. Human is standing on the kitchen floor.  2. Human moves to the fridge, opens it, and picks up the fruit.  3. Human moves to the countertop, and puts the fruit on it.

4. Human asks for help:  **Can you hand me that one on the table please?**

5. Robot starts to work, with possible options for '**that one**':

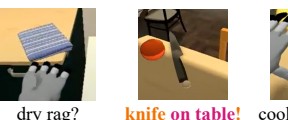 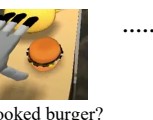

dry rag?        **knife on table!**        cooked burger?        ......

Figure 1: An example HandMeThat task, rendered in images. A robot agent observes a sequence of actions performed by the human (step 1-3), and receives a quest (step 4). The robot needs to interpret the natural language quest based on physical and social context, and select the relevant object from the environment: the knife on the table in this case. Currently, the HandMeThat benchmark is released as a text-only environment.

following in physical and social environments. We set up the environment using an internal symbolic representation-based simulator, so that we can generate human trajectories and instructions following automated pipelines and build automated evaluation protocols.

Each episode in HandMeThat contains two stages. In the first stage, the robot will be watching a human-like agent taking actions towards her internal goal (which is unrevealed to the robot). At the end of the first stage, we assume that the human needs help from the robot. Therefore, the robot receives a (possibly ambiguous) language instruction, which is essentially a subgoal for the human. In the second stage, the robot takes actions in the environment to accomplish this subgoal. We evaluate the robot's performance by his action costs during the second stage and whether his actions accomplish human's subgoal.

HandMeThat contains a diverse set of physical (the objects in the scene) and social (the internal goal of human) information. Specifically, each HandMeThat scene contains 14 locations and typically more than 200 movable objects, which induces a large set of possible actions. The human's internal goal is instantiated from a distribution derived from BEHAVIOR-100 [7]. This brings us two advantages. First, since BEHAVIOR tasks are manually annotated, they follow the natural distribution of human household tasks. Second, due to the compositional nature of BEHAVIOR-100, the space of possible goal specifications is enormous. In particular, using the templates we extracted from BEHAVIOR-100, we can instantiate more than 300k distinct tasks. Such diversity introduces important challenges for goal recognition, language understanding, and embodied interaction.

The key challenge in HandMeThat tasks is the recognition of human's subgoal from her historical actions and ambiguous instructions. This resembles three important challenges: recognition of human goals, pragmatic reasoning of natural language, and planning. However, HandMeThat is not a simple *ensemble* of these three challenges. In goal recognition, the robot needs to consider both human's historical actions as well as the subgoal specified in human utterance. Similarly, the context for pragmatic reasoning consists of both the physical environment and human's internal goals. Furthermore, since our environment is partially observable, the robot can gather additional information through exploration in order to help with goal recognition and pragmatic reasoning. HandMeThat integrates these naturally-occurring challenges and serves as a holistic benchmark.

In this paper, we implement a textual interface to render physical scenes and allow the robot agent to use natural language commands to interact with the human and objects. Rendering in the textual interface bypass the difficulties in visual perception and recognition, allowing us to focus on pragmatic instruction reasoning problem. We formulate the learning problem using reinforcement learning, where the robot receives textual inputs describing the scene and the human actions, and generates

| | Object Interaction | Goal Space (#Templates) | Common-Sense Goal Prior | Instruction Interpretation | Social Reasoning | Pragmatic Inference | Collaborative Task Completion |
|---|---|---|---|---|---|---|---|
| BEHAVIOR [7] | ✓ | 100 (100) | ✓ | ✗ | ✗ | ✗ | ✗ |
| ALFRED [6] | ✓ | 7,000 (7) | ✗ | ✓ | ✗ | ✗ | ✗ |
| Watch-and-Help [8] | ✓ | 5 (5) | ✗ | ✗ | ✓ | ✗ | ✓ |
| CerealBar [9] | ✓ | 1202* | ✗ | ✓ | ✓ | ✗ | ✓ |
| ALFworld [10] | ✓ | 7,000 (7) | ✗ | ✓ | ✗ | ✗ | ✗ |
| DialFRED [11] | ✓ | 36,912 (25)* | ✗ | ✓ | ✗ | ✗ | ✓ |
| TEACh [12] | ✓ | 7,000 (7) | ✗ | ✓ | ✓ | ✗ | ✓ |
| Two Body [13] | ✓ | 1 (1) | ✗ | ✗ | ✗ | ✗ | ✓ |
| TextWorld [14] | ✓ | — | ✗ | ✓ | ✗ | ✗ | ✗ |
| LIGHT [15] | ✗ | 10,777* | ✗ | ✓ | ✓ | ✗ | ✗ |
| SCONE [16, 17] | ✗ | 13,951* | ✗ | ✓ | ✗ | ✓ | ✗ |
| HandMeThat (ours) | ✓ | >300k (69) | ✓ | ✓ | ✓ | ✓ | ✓ |

Table 1: Comparison between HandMeThat and other related benchmarks. *: indicates the number of instructions or dialogues in that dataset, in contrast to explicit goals/tasks. The numbers in the parenthesis in the third column represents the number of goal templates.

natural language commands to accomplish the subgoal specified by the human. For baselines, we compare two groups of baselines. The first set contains a random agent and a heuristics-based agent. The second set contains neural network baselines that are trained with offline and online reinforcement learning algorithms. All learning-based baselines show less than 20% success rate on our held-out test episodes, suggesting significant room for improvements.

## 2 Related Work

Table 1 summarizes the comparison between HandMeThat and other text-based and vision-based benchmarks in similar domains. Although HandMeThat is only rendered in texts currently, simplifying the perception and recognition problem, we believe that many other aspects we emphasize are critical challenges and it is worth comparing HandMeThat with many visual benchmarks.

**Household manipulation tasks.** Robotic manipulation in household environments is an important challenge because it calls for combined research of navigation, object manipulation, and language use. Thus, many household environment simulators and platforms [18, 19, 20, 21] have been built. As a representative, BEHAVIOR-100 [7] (built on iGibson 2.0 [22]) is a physics-based simulator and the only one that contains human-annotated tasks, which reflects a real-world distribution of household activities. In this paper, we leverage the state representations and task distributions collected by BEHAVIOR-100 to study human-robot communication. The task distribution can be viewed as a commonsense prior of human goals and intentions: e.g., human uses knife to cut fruit (in contrast to hammers) and jars to store them (in contrast to trash bins). By contrast, the original BEHAVIOR-100 does not involve any language communication. The closest benchmark to our work is ALFRED [6] and its successor ALFWorld [10] (built on AI2-THOR [23]). Both benchmarks use natural language instructions to set up the tasks. However, their task distribution is not diverse: ALFRED has 7,000 different goals in compositional formulas, instantiated from only 7 templates, and they do not follow any real-world task distribution. Furthermore, both ALFRED and ALFWorld do not consider pragmatic reasoning and human actions and goals.

**Goal recognition and social reasoning.** Our task formulation is closely related to the literature on goal recognition: inferring the goal of other agents based on their historical actions [24, 25, 26, 27, 28, 29, 30, 31]. The most prevalent assumption is the principle of rationality: agents should make (approximately) optimal decisions to achieve their goals, given their beliefs [1, 2]. Similar to existing work on goal recognition from unstructured data [32], in HandMeThat, the robot does not assume access to domain knowledge. Furthermore, our benchmark HandMeThat makes an important extension to the standard setups of goal recognition: besides human actions, HandMeThat considers (possibly ambiguous) human instructions that set subgoals for the robot. The objective of the robot is not to fully recover the human's goal, but the subgoal set by the human.

Understanding human intentions in embodied environments has been studied in other works. Watch-and-Help [8] introduces a non-language goal inference task based on the VirtualHome environment [18]. One critical drawback of their environment is that the goal space is relatively small, and no

language interpretation is involved. CerealBar [9] is another dataset for robotic instruction following in a collaborative environment. However, their environment does not reflect real-world priors of goals. LIGHT [15] sets up a textual platform for understanding actions and emotes within natural language dialogues, which involves the interpretation of human's internal ideas by focusing on background knowledge such as backstory and personality. Our benchmark, on the contrary, aims at reasoning the internal goal within the planning domain.

**Pragmatic inference.** Our work is closely related to the extensive literature on pragmatic inference—the study of context contributes to meaning. This idea is widely applied to referring expression generation tasks [33, 34] in visual scenes. Our benchmark, similarly, considers human utterance generation for subgoals and considers both physical and social contexts. Related to HandMeThat, there is work on integrating pragmatic reasoning and instruction following tasks. Fried et al. [16] proposed a unified pragmatic model for generating and following instructions, which applies pragmatic inference on the navigation dataset SAIL [35] and the semantic parsing dataset SCONE [17]. Both tasks focus on modeling the textual contexts (e.g., the previous instructions in a dialog). By contrast, in this paper, we focus on understanding human actions in physical and social environments.

**Collaborative Communication.** In Two Body Problem [13], two agents can communicate in both explicit (through message) and implicit (through perception) ways, in order to efficiently finish a single given task. DialFRED [11], an extension to ALFRED allows agent to actively ask questions to humans for helpful information. However, the language in DialFRED has no ambiguity, and the questioning is only for information seeking procedure instead of ambiguity resolution. TEACh [12] and CerealBar [9] introduce collaborative tasks where a *Commander* receives a given task and a *Follower* interacts with the environment. In their work, the objective of the *Commander* is to accurately describe the task in language to the *Follower*. By contrast, our benchmark focuses explicitly on the trade-off between informativeness and communication cost.

**Text-based reinforcement learning.** We build a textual interface for HandMeThat based on gym environment [36], which have been used by adventure game environments such as Textworld [14] and Jericho [37]. In contrast to our benchmark HandMeThat, these environments do not involve goal-conditioned learning, and there is no human-robot communication and interactions.

## 3   The HandMeThat Benchmark

The data generation and evaluation of HandMeThat are based on a symbolic representation-based simulator. In this section, we will borrow notations and concepts from classical planning domains [38, 39] to formalize our environment. A domain $\Xi$ is composed of a state space $\mathcal{S}$, an action space $\mathcal{A}$, and a transition function $\gamma$. Each state $s \in \mathcal{S}$ is represented as an object-centric representation $s = \langle \mathcal{U}, \mathcal{V} \rangle$. $\mathcal{U}$ is the universe of objects in the scene. $\mathcal{V}$ is a finite set of state variables. All state variables $v \in \mathcal{V}$ considered in this paper are binary-valued. They are either unary variables that describe the attributes of each object (e.g., `sliced(apple#0)`, `dusty(box#1)`), or binary variables representing the spatial relations between objects (e.g., `in(apple#1, box#2)`). For convenience, we will use $p, r$ to denote the set of attributes and relations separately. In this paper, we only consider two spatial relationships: *on* and *in*. All object categories and states are inherited from BEHAVIOR-100.

In HandMeThat, the action space $\mathcal{A} = \mathcal{A}_h \cup \mathcal{A}_r$ is composed of human actions $\mathcal{A}_h$ and robot actions $\mathcal{A}_r$. Each action $a \in \mathcal{A}$ can be represented as $a = \langle \hat{a}, O_{arg} \rangle$, where $\hat{a}$ is an action schema [40] and $O_{arg}$ is a tuple of object arguments $o_1, \ldots, o_k \in \mathcal{U}$. For example: `robot-open(cabinet)` means "robot opens the cabinet," and `robot-slice-with-on(human, apple, knife, table)` means "the robot slices an apple with knife on the table." In STRIPS planning literature, each action $a$ can be considered as a grounded operator, characterized by preconditions and effects, which are logical formula defined over state variables and assignments to state variables, respectively. We leave the detailed definitions of these operators to the supplementary material. As an example, the action `robot-open(cabinet)` changes the `is-open` property of the cabinet.

The set of available actions (grounded operators) immediately induces a deterministic and discrete transition function: $\gamma : \mathcal{S} \times \mathcal{A} \to \mathcal{S}$. $\gamma(s, a)$ computes the outcome state after taking action $a \in \mathcal{A}$ at state $s \in \mathcal{S}$. Meanwhile, we define the cost function $\mathcal{C} : \mathcal{S} \times \mathcal{A} \to \mathbb{R}$, which computes the effort $\mathcal{C}(s, a)$ in performing $a$ at state $s$. For simplicity, throughout the paper we assume actions have unit costs. The details of our state representation, action schemas, and action costs can be found in the supplementary material.

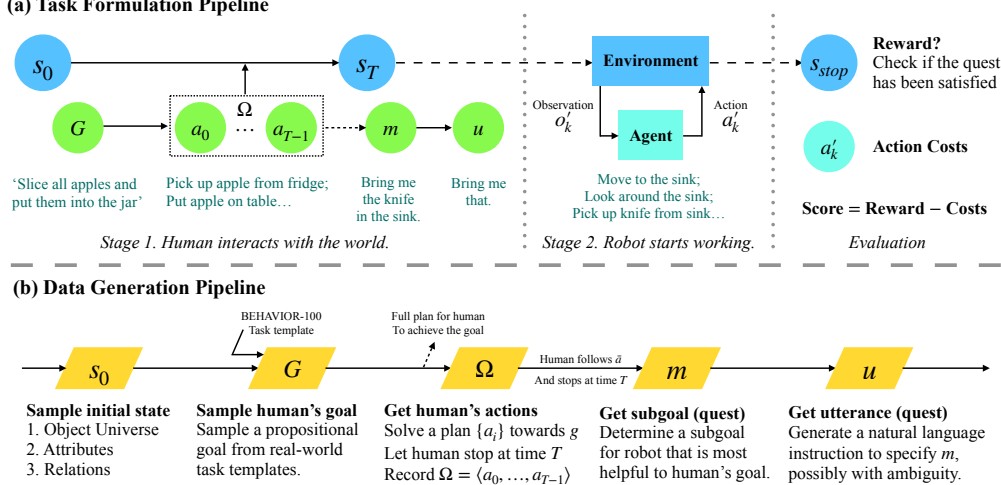

Figure 2: (a) A pipeline for HandMeThat task formulation. Stage 1: Human takes $T$ steps from initial state $s_0$ towards a goal $G$, giving a trajectory $\Omega$. At state $s_T$, she generates a subgoal $m$ for robot and utters it as $u$. Stage 2: Robot perceives and acts in the world, following the human's instruction. Evaluation: When the robot stops, we check if human's quest has been satisfied, and count robot's action costs to give a final score. (b) A pipeline for HandMeThat data generation. We first sample initial state $s_0$, human's goal $G$, and solve a plan for human to execute. At a randomly sampled step $T$, the human stops and generates a subgoal, including both the internal $m$ and utterance $u$.

Based on the basic definitions of states, actions, and transition functions, we formally define each HandMeThat episode as a tuple $\langle s_0, G, \Omega, s_T, m, u \rangle$. As shown in Fig. 2a, each episode consists of two stage. In the first stage, the human agent takes $T$ steps from the initial state $s_0$ towards goal $G$. The trajectory $\Omega$ is the sequence of human actions, and $s_T$ is the state reached by executing $\Omega$. Then, the human determines a subgoal $m$ in her mind and specifies it through the utterance $u$. In the second stage, the robot observes $\langle \Omega, s_T, u \rangle$, and interacts with the environment. The performance of the robot will be evaluated by whether his actions accomplish the subgoal $m$ and the total costs of his actions.

Fig. 2b presents the generation pipeline of HandMeThat episodes. We start from randomly sampling the initial world state $s_0$ and the goal $G$ for humans. Second, we generate the human trajectory $\Omega$ assuming that human takes an optimal plan towards the goal (Section 3.1). Next, we generate the subgoal $m$ and utterance $u$ (Section 3.2). The key assumption is that the subgoal $m$ is an *useful* subgoal towards $G$, while $u$ is generated with a Rational Speech Acts (RSA) model.

## 3.1 Intial States, Goals, and Human Trajectories

The initial state $s_0$ in each HandMeThat episode is randomly generated by sampling the number of objects of each object category and then their attributes and spatial relationships. As a result, each scene contains more than 200 entities with diverse attributes, which resembles a typical real household environment. We show one example of generated scenes in Fig. 3a.

Based on the object-centric state representation, a goal $G$ can be defined as a first-order-logic formula over objects in $\mathcal{U}$. For example, `is-open(cabinet)`$\wedge$`on(apple#0, table)`. We say a state $s$ satisfies $G$ if the formula $G$ evaluates to true at $s$. In each episode, the human generates an internal goal $G$, which is unrevealed to the agent. The goal space $\mathcal{G}$ is derived from human-annotated household tasks in BEHAVIOR-100, represented using templates of first-order logic statements. For example, the "bottling fruit" task (storing sliced fruits) can be formalized as the template shown in Fig. 3b, and can be instantiated by replacing blanks with concrete object properties.

Given the domain $\Xi$, the sampled goal $G$, and the initial state $s_0$, we consider a planning task $\Pi = \langle \Xi, s_0, G \rangle$. A solution to a planning task is a sequence of actions $\pi$ that reaches a goal state $G$ starting from the initial state $s_0$ by following the transitions defined in $\Xi$. We first use a planner to generate a trajectory $\pi = \langle a_0, \ldots, a_n \rangle$ that accomplishes the goal. Next, to simulate the scenario

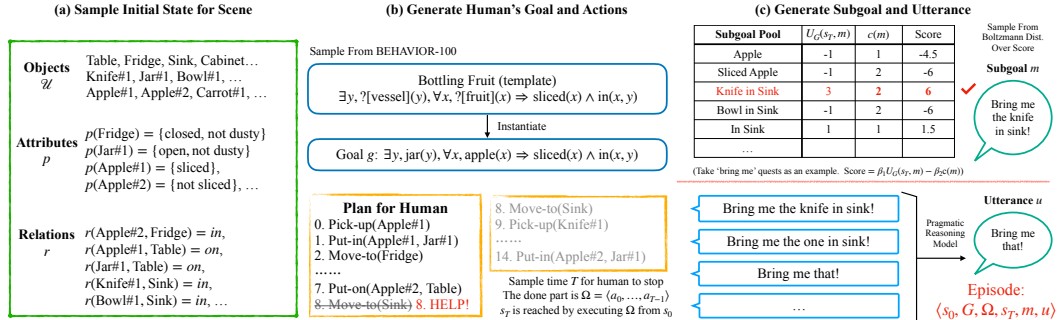

Figure 3: Example of our generated scene, goal, and quest. (a) The sampled initial state. (b) The sampled internal goal from BEHAVIOR-100 templates and the plan. (c) The generated quest in mind based on human's utility and a corresponding utterance generated using the Rational Speech Acts (RSA) model [41].

where the human asks for help, we randomly truncate the trajectory into $T$ steps. The robot agent observes human actions $\Omega = \langle a_0, \ldots, a_{T-1} \rangle$ and the final state $s_T$.

### 3.2 Subgoals and Utterances

In this paper, we consider three types of subgoals: "bring-me" (the robot should deliver an object to the human), "move-to" (the robot should move an object to a designated location), and "change-state" (the robot should change the state of an object, such as cleaning). Instead of requesting a specific object, the object and target location referred in $m$ is specified using object attributes and relations. That is, instead of specifying "Bring me the plate with ID 93," the subgoal might be "Bring me a large ceramic one," or "Bring me the plate on the dining table." This resembles the real-world use of natural language.

Formally, we call $m$ a *lifted* subgoal. Before explaining that, we define a *grounded* subgoal $mg$ to be a propositional logic formula over state variables $\mathcal{V}$, e.g., `human-holding(plate#93)`. Intuitively, a lifted subgoal replaces each concrete object with a dictionary specifier $d$. For example, `plate#93` is replaced by a specification dictionary `{size:large, material:ceramic}`. The specification dictionary can be translated into a formal first-order-logic formula over objects in $\mathcal{U}$, as $\exists x.$ `human-holding`$(x) \wedge$ `size-large`$(x) \wedge$ `material-ceramic`$(x)$

For each lifted subgoal $m$, we use $A(m)$ to denote the set of all possible grounded subgoals that satisfy $m$. Formally, $A(m) = \{mg | \forall s \in \mathcal{S}.mg(s) \to m(s)\}$, where $mg(s), m(s)$ denote the results of evaluating $mg, m$ on state $s$, respectively.

**Subgoal generation.** The subgoal generation process is based on the assumption that the human chooses a lifted subgoal that maximizes her internal reward. Such internal reward is composed of two parts: the utility of the subgoal and the complexity of the subgoal.

To quantify the utility of the subgoal, we first define a set of helpful functions. Based on $\gamma$ and $\mathcal{C}$, we denote the optimal (goal-conditional) policy $\pi_G^*(s) : \mathcal{S} \to \mathcal{A}$ which computes the first action starting from $s$ to achieve the goal $G$. Furthermore, we define cost-to-go $V_G^*(s)$ as the total cost following $\pi_G^*(s)$ before achieving $G$.

Next, we define a grounded-subgoal transition function $\gamma_{\text{subgoal}}(s, mg)$. Specifically, let $s_0$ be the current state, $a_0, s_1, a_1, \cdots, a_{k-1}, s_k$ be the unrolling of the optimal policy $\pi_{mg}^*(\cdot)$: $a_i = \pi_{mg}^*(s_i)$, $s_{i+1} = \gamma(s_i, a_i)$. We define $\gamma_{\text{subgoal}}(s_0, mg) = s_k$. Intuitively, this function computes the landing state following the optimal policy $\pi^*$ towards a grounded subgoal $mg$. Thus, we can define the utility of a subgoal as:

$$U_G(s_T, m) = V_G^*(s_T) - \frac{1}{|A(m)|} \sum_{mg \in A(m)} \left[ V_G^* \left( \gamma_{\text{subgoal}}(s_T, mg) \right) \right]$$

where $s_T$ is the state that the human generates the quest. This function quantifies the expected difference between the current cost-to-go and the cost-to-go after the agent accomplishes the subgoal

*mg*, assuming *mg* is uniformly sampled from $A(m)$. When $V_G^*(\gamma_{\text{subgoal}}(s_T, mg)) < V_G^*(s_T)$, we say that *mg* is *useful* for $G$. We further define $A(G)$ to be the set of all *useful* grounded subgoals for $G$.

We define the complexity of a subgoal $c(m)$ as the number of specifiers in $m$. For example, the subgoal "bring-me(in sink)" has only one specifier, and thus its cost is 1. The human chooses a subgoal $m$ based on the following distribution:

$$P(m|s_T, G) \propto \exp[\beta_1 U_G(s_T, m) - \beta_2 c(m)],$$

where $\beta_1 = 3, \beta_2 = 1.5$ are inverse temperature constants of the Boltzmann distribution.

**Utterance generation.** Our utterance generation process follows the Rational Speech Acts (RSA) models. It generates utterance $u$ given the underlying meaning (i.e., the subgoal) $m$. The utterance $u$ has the same format as $m$, but the object specification dictionaries are less constrained. For example, instead of specifying both the size and the material, the utterance may only contain material specifications {material:ceramic} (i.e., "Bring me a ceramic one.") In the extreme case, the object specifier can be empty, corresponding to the natural language "Bring me that." For convenience, we say $m \subseteq u$ if and only if $A(m) \subseteq A(u)$.

The intuition behind the generation of $u$ follows the RSA model, where the speaker (the human) considers a rational listener (the robot). Formally, the RSA model iteratively defines a sequence of distributions:

$$P_{L_0}(m|u) \propto l(u, m)P(m);$$
$$P_{S_k}(u|m) \propto \exp[\alpha U_{S_k}(u, m)]; \quad P_{L_k}(m|u) \propto P_{S_k}(u|m)P(m).$$

Here, for $k \geq 1$, $U_{S_k}$ is a utility function computed as $U_{S_k}(u, m) = \log P_{L_{k-1}}(m|u) - \alpha' c(u)$. $\alpha = 2, \alpha' = 1$ are temperature constants and cost weights. The literal meaning of a utterance $l(u, m)$ is defined as $\mathbf{1}[A(m) \subseteq A(u)]$, where $\mathbf{1}$ is the indicator function. Intuitively, if $m$ is a finer-grained specification than $u$, $m$ satisfies $u$. $P(m)$ is the prior distribution of possible meaning $m$, which is defined as $P(m) = P(m|s_T, G)$—essentially assuming that the listener has correctly recognized the internal $G$ of the human. We use the same cost function $c(u)$ as $c(m)$, which counts the number of specifiers. In HandMeThat, we use a non-trivial $k = 10$ and uses the distribution $P_{S_k}(u|m)$ to sample an utterance $u$ given the meaning $m$. The sampled utterance $u$ is translated into natural language following templates detailed in our supplementary material. The choice of hyperparameters $\beta_1, \beta_2, \alpha, \alpha', k$ is also discussed in supplementary materials.

Fig. 3c shows a concrete example for utterance generation. When the human asks for help at time $T = 8$, the remaining part towards her goal is to get a knife and slice the apple. Considering the pool of all possible subgoals, the ones that are related to "delivering the knife" have a high utility. Based on the sampled subgoal $m$ "Bring me the knife in the sink," we list all possible utterances $u$ that satisfy $m$, by removing a portion of the specifiers ("in-sink" and "is-knife" in this example) and use the RSA model to generate the utterance.

### 3.3 Hardness Levels

The HandMeThat benchmark holistically evaluates language grounding, goal inference, and pragmatic reasoning. To systematically disentangle these challenges, we split HandMeThat into four hardness levels, and the gaps between levels correspond to different challenges. Recall that $A(G)$ denotes the set of all *useful* grounded subgoals for goal $G$. $A(m)$ and $A(u)$ are the sets of all possible grounded subgoals for the lifted subgoal $m$ and the utterance $u$. Finally, we consider the subgoal derived from pragmatic reasoning. Specifically, denote $r = \arg\max L_k(m|u)$ (i.e., the most probable subgoals following RSA given $u$). Denote its corresponding grounding set as $A(r)$. Directly from the definitions, we have $A(m) \subseteq A(G), A(m) \subseteq A(u), A(r) \subseteq A(u)$. The four hardness levels are:

**Level 1**: $A(m) = A(u)$. The utterance has no ambiguity. In this case, the instruction understanding task is a pure grounding task: the agent only needs to select the object that satisfies the specification.

**Level 2**: $A(m) = A(u) \cap A(G)$. The second level requires social reasoning: the robot can successfully accomplish the task if it can both ground $u$ and infer the human goal $G$ from observations.

**Level 3**: $A(m) = A(r)$. The third level requires all reasoning capabilities combined: the agent need to infer the human goal $G$. Next, it should make pragmatic reasoning based on $G$ and $u$ to derive $r$.

**Level 4**: $A(m) \subset A(r)$. In this case, the human utterance $u$ is inherently ambiguous and can not be resolved even with all reasoning capabilities. In this case, further information gathering is needed.

### 3.4 Text-Based Interactive Interface

In this paper, we construct a text-based environment for HandMeThat. We implement the environment based on the gym environment interface [36]. We present an running example of our gym environment in supplementary materials. The initial observation contains the trajectory of human and the language instruction $u$. Every step, the agent can execute robot actions, including moving, examining, and manipulating objects (pick-and-place, heat, etc.). After each action, the agent will receive a new observation containing the object state changes. Each action has the cost $\mathcal{C}(s, a)$ which is computed using the internal state. Meanwhile, after each action, the environment will check if the subgoal set by the human has been accomplished. When the agent succeeds, it will receive a reward of 100. We set the maximum episode length to 40. Thus, the agent will fail after taking 40 actions.

We consider two environmental settings: fully-observable and partially-observable. Concretely, in the fully-observable setting, the initial observation contains the information of all objects in the environment. By contrast, in the partially-observable setting, the robot can only see the objects at his current location. For receptacles, the robot needs to explicitly open the receptacle to investigate the objects inside. Our text-based environment has a vocabulary of size 250. The average token length of the observation is 860 in the fully observable setting and 140 for the partially observable setting.

### 3.5 Challenges in HandMeThat Tasks

The HandMeThat resembles three important and co-occurring challenges: goal inference, pragmatic reasoning, and planning. In this section, we briefly discuss the interplay between them and the new challenges set up by HandMeThat. First, unlike most relevant literature on goal recognition, which focuses on inferring goals from human actions, HandMeThat additionally considers the ambiguous instructions from humans. Furthermore, the target of the task is not to fully recover the internal goal $G$ of the human, but the subgoal $m$ set by instructions, by assuming that $m$ is a useful subgoal towards $G$. Another important challenge in HandMeThat is that in real-world deployment of robots, the robot needs to *learn from experience* human preferences, such as object placements and dominant hands. A promising direction is to integrate learning algorithms with goal recognition algorithms. For example, Sohrabi et al. [42] discussed integrating external probability distributions into planning domains for goal recognition. Second, in the pragmatic reasoning tasks of HandMeThat, the robot needs to consider both the physical states of objects but also the goals and potential subgoals of humans. Thus, HandMeThat can serve as a benchmark towards building language understanding models grounded on not only physical states but also human actions and goals. Finally, as a partially-observable environment, the robot can and should take actions to gather additional information to facilitate his goal recognition and pragmatic reasoning. In our environment, this primarily involves searching for relevant objects for goal recognition and pragmatic reasoning.

## 4 Experiments

We evaluate two sets of methods on HandMeThat. The first set of models contains a random agent and a heuristic-based agent. The second set is neural network-based agents trained with offline and online reinforcement learning algorithms.

### 4.1 Model Details

**Hand-coded models.** The first model (Random) is an agent that randomly selects a valid action at each time step. The second model (Heuristic) is an agent that heuristically repeats the previous human actions. To be more specific, this model has access to all object states and the underlying logic formula of the utterance. Therefore, it is only applicable in the fully-observable setting. Upon receiving the instruction, the agent generates all possible groundings of instruction and then compares them to the observed human trajectory. The key heuristic of this model is that: human tends to quest for objects that are in the same categories as the previously manipulated ones. The Heuristic model guesses the grounding of the objects in the utterance based on this heuristic.

**Neural network models.** We evaluate two neural network baselines. The first model (Seq2Seq) is based on the sequence-to-sequence model [43] for language modeling, trained with behavior cloning [44] on expert demonstrations. The expert demonstrations are generated by applying greedy

| Model | Partially Observable | | | |
|---|---|---|---|---|
| | Level 1 | Level 2 | Level 3 | Level 4 |
| Human | (76%, 5.1) | (52%, 4.8) | (20%, 5.8) | (8%, 5.0) |
| +fully* | (92%, 4.4) | (80%, 4.8) | (36%, 4.3) | (16%, 4.5) |
| Heuristic* | 88.9 (94.9%, 4.2) | -1.9 (28.1%, 4.4) | -23.7 (12.0%, 4.3) | -15.9 (17.8%, 4.4) |
| Random | -40.0 (0.0%, N/A) | -39.5 (0.4%, 16.0) | -40.0 (0.0%, N/A) | -40.0 (0.0%, N/A) |
| Seq2Seq | -5.4 (25.5%, 4.2) | -25.8 (10.4%, 4.1) | -34.7 (3.9%, 4.1) | -32.8 (5.3%, 4.1) |
| +goal | -11.4 (21.0%, 4.1) | -28.8 (8.2%, 4.1) | -30.0 (7.4%, 4.2) | -32.5 (5.5%, 4.0) |
| +subgoal | -10.1 (22.0%, 4.1) | -18.7 (15.7%, 4.1) | -24.5 (11.4%, 4.1) | -21.9 (13.4%, 4.1) |
| DRRN | -40.0 (0.0%, N/A) | -40.0 (0.0%, N/A). | -40.0 (0.0%, N/A) | -40.0 (0.0%, N/A) |
| +offline | -40.0 (0.0%, N/A) | -40.0 (0.0%, N/A). | -40.0 (0.0%, N/A) | -40.0 (0.0%, N/A) |

Table 2: Experiment results in the partially observable setting. Each model is evaluated on 4 hardness levels with 3 metrics: 1) the average score, 2) the success rate, and 3) the average number of moves in successful episodes. *: indicates the results from fully-observable setting for reference.

best-first search with FF heuristic [45]. We apply the architecture of ALFWorld Seq2Seq baseline, where the hidden representation of observation strings is obtained by using task description as attention. As a reference, we also provide the performance of Seq2Seq model given extra oracle (goal or subgoal of the human agent in first-order logic format). The second model (DRRN) is presented in Jericho [37], which is a choice-based text-game agent based on Deep Reinforcement Relevance Network (DRRN) [46]. It learns a Q function for possible state-action pairs. We also discuss an offline variation of DRRN. Specifically, instead of actively collecting environmental trajectories based on the current policy, we train the Q function network with expert demonstration trajectories.

## 4.2 Results

The performance of learning models in partially-observable setting is shown in Table 2. For reference, we also present the performance of human players and the heuristic model. The other experiment results are included in supplementary materials. We consider three evaluation metrics: 1) the average score of the model; 2) the success rate that the model achieves the goal within limited steps (40); 3) the average number of moves of successful episodes. Scores are averaged on 1,000 episodes.

The Heuristic model is the best performing model. Its high success rate on Level 1 is mainly because the utterances in level 1 are unambiguous. Since the Heuristic model assumes groundtruth full information about object states and the underlying symbolic representation of utterances, it can successfully interact with the objects. It also has access to a planner that resolves plans for Pick-and-Place tasks. To fairly compare it with the other baselines, we only allow the model to perform in a one-trial manner, i.e., the model could only guess once about the specified subgoal and then follow the plan towards it. However, when the level gets harder, its success rate drops, and it is increasingly harder for our simple heuristic to select the correct object.

Among the learning-based models, the Seq2Seq model performs decently well on both fully- and partially-observable settings. Its performance also gets worse when the hardness level increases. Note that the average number of moves when Seq2Seq succeeds is low and stable. It is possibly because the model has learned a fixed template for many pick-and-place tasks that can be accomplished in 4 or 5 steps. There's a large gap between Seq2Seq and the Heuristic model. Apart from the extra information provided to the Heuristic model, there are several other challenges for the Seq2Seq model. First, human goal space is enormous. Thus, a similar scene and instruction may not correspond to the same goal. Directly learning a mapping from observations to actions can be prone to spurious contexts. Interestingly, for level 1 and 2, the Seq2Seq model shows slightly better performance in the partially-observable setting, though the expert demonstration is the same. We attribute this to the network capacity issue. Compared with the partially-observable model, the fully-observable model uses a separate encoder for the full observation strings. Since we keep the model size to the same, the partially-observable model has doubled the parameters for encoding task-relevant information such as the objects at the current location. When the models are trained with extra oracle information (goal or subgoal of human), all the performances except level 1 significantly improve. This highlights that inferring human goals and subgoals is helpful in resolving ambiguous instructions.

The reinforcement learning-based DRRN model, by contrast, totally fails in both settings, yielding a similar performance as the Random agent. This is primarily due to the large action space for the model. At each step, there are 15 to 30 valid actions, and the agent gets a sparse reward only when it successfully accomplishes the task. Therefore, in practice, we see that the random exploration strategy in the beginning stage of DRRN's training struggles to find positive rewards.

As a reference, we present human performance on HandMeThat. We collect human performance on each hardness level. We first introduce human subjects to the benchmark settings including possible locations, objects, and tasks. Next, human subjects interact with our textual interface. For both fully- and partially-observable settings, we collect 25 episodes of human performance for each hardness level. Human subjects are only allowed to choose one particular object to manipulate, and stop immediately after execution no matter if the goal is reached. The results show that our human subjects can perform well on tasks in level 1 and 2. In level 3, the inference of speaker intention gets more difficult. As a result, the performance drops significantly, but still outperforms our learning baselines. Tasks in level 4 are intrinsically under-specified. Thus, the human subjects perform worse than level 3. Note that our environment supports robots asking additional clarification questions to the simulated human in order to resolve ambiguities. However, since no current learning models are not capable of actively asking questions, for a fair comparison, we do not enable this option in human experiments.

In general, the performance of all baselines is low on our HandMeThat benchmark, especially for level 3 and level 4 tasks. There are two main reasons for this observation. First, there exists a large number of candidate objects in the environment. In particular, when there are multiple objects with similar categories (e.g., different kinds of drinks), it can be hard for models to select the objects that are relevant to human's goals and ambiguous instructions. Second, a portion of the level 3 and level 4 tasks are "intrinsically" unsolvable. That is, the information in human's historical actions and instructions may not be sufficient to accurately select the relevant objects. This is a desired feature for level 4 tasks, where further information gathering is required (see our supplementary for more discussions on the extension of the dataset). For level 3 tasks, such information insufficiency occurs as an artifact of the Rational Speech-Acts model. Specifically, we are using the groundtruth goal specification when computing listener's prior. Theoretically, we should replace this groundtruth information with a distribution of possible human goals that can be inferred from the historical actions and instructions, for example, by leveraging inverse planning algorithms [25]. However, this requires an intractable computation due to the large size of our goal space. Therefore, we choose to implement the current RSA computation based on groundtruth goal information.

## 5   Conclusion

We introduce HandMeThat, a benchmark for evaluating instruction understanding and following within physical and social contexts. HandMeThat requires the ability to resolve possible *ambiguities* in human instructions based on both the physical states of scene objects and human's internal long-term goal. We present a textual interface for HandMeThat and evaluate several baselines on it. The experiment results suggest that HandMeThat introduces important challenges for human-robot interaction models. We hope that HandMeThat will motivate the development of more robust systems that can accomplish Human-AI communication in complex physical and social environments. Our benchmark focuses on resolving ambiguous instruction, but there exist more uncertainty issues in real-world Human-Robot Interaction (HRI) tasks (e.g. non-verbal communication [47]). Thus, in future work, we may render HandMeThat into a visual interface and add back the other challenges within the visual domain. Another exciting challenge will be to extend the current HandMeThat to multi-round human-robot communications, essentially forming a physically and socially grounded dialog. Future research may consider joint pragmatics reasoning of physical, social, and textual contexts, and information gathering through language.

**Acknowledgements.** We thank the anonymous reviewers for their constructive comments and suggestions. We thank our friends and colleagues for participating in the human experiments and providing helpful feedback. This work is in part supported by ONR MURI (N00014-13-1-0333), the Center for Brain, Minds, and Machines (CBMM, funded by NSF STC award CCF-1231216), the MIT Quest for Intelligence, and MIT–IBM AI Lab. Any opinions, findings, and conclusions or recommendations expressed in this material are those of the authors and do not necessarily reflect the views of our sponsors.

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
