# OpenReview forum: "HandMeThat: Human-Robot Communication in Physical and Social Environments"
_NeurIPS.cc/2022/Track/Datasets_and_Benchmarks — NeurIPS 2022 Datasets and Benchmarks _

### Official Review · Reviewer_ydPb · 2022-07-20
**A benchmark based on textual interface for human-robot communication**

**Rating:** 6
**Confidence:** 4

**Strengths:**

Strengths:
- The proposed benchmark looks at an important problem of service robots assisting humans with their goals. To infer human intents, the paper proposes a specific task which involves agent reasoning based on implicit observations and explicit communication signals.
- The paper is generally well written and technically sound. The authors provide sufficient details in the paper to help readers understand how the benchmark is designed and how the dataset is generated.


**Weaknesses:**

Weaknesses:
- Most importantly, I believe the text-based environment over-simplifies the problem. Many factors that prevent the agent from predicting human's goals and intention mainly exist in the visual domain (e.g. occlusion, detecting interacted objects). The textual setting fails to consider these factors.
- Some task settings are not justified and the application is not clear. I understand that due to communication cost, the human's utterance can not completely specify the subgoal. However, during human communication, most of the time the speaker should be in favor of accuracy rather than brevity. "Hand me that" is a rather rare case for human communicators, only occurs when the common ground is clearly established. Since adding a few words can greatly reduce the search space of the listener, It is not clear whether such a tradeoff are necessary and meaningful in reality.
- In terms of literature, some relevant references are missing and the comparison is baised (see below).


**Additional Feedback:**

N/A

**Clarity:**

The paper can benefit from a revision to improve clarity:
- Since the proposed benchmark is mainly based on textual environment, the images displayed in figure 1 cause misunderstandings for the readers. The authors should consider adding a note in the caption for clarification.
- More explanations are needed for section 3.3. In line 200, the authors state the assumption that the listener knows the internal goal of the human. Isn't the goal inference an essential part of the task?
- The authors should explain how the values of hyperparameters (alpha, beta, k) are chosen and how they affect the results.


**Correctness:**

Correctness:
- I understand the cost of subgoals are incoporated in the utterance generation to reflect a human tradeoff between communication cost and information precision. However, the cost term also exists during the meaning generation. How does an accurate sub-goal specification reduce human's internal reward?
- It seems the values in the table of fig3(c) do not align with the equation in the main text.



**Documentation:**

I believe the authors have released code and dataset with detailed instructions to reproduce the results.

**Ethics:**

There is no ethical issue from my perspective.



**Relation To Prior Work:**

- Some relevant references are missing. The authors are recommended to check and add the references below:

Gao, X., Gao, Q., Gong, R., Lin, K., Thattai, G., & Sukhatme, G. S. (2022). Dialfred: Dialogue-enabled agents for embodied instruction following. arXiv preprint arXiv:2202.13330.

Padmakumar, A., Thomason, J., Shrivastava, A., Lange, P., Narayan-Chen, A., Gella, S., ... & Hakkani-Tur, D. (2022, June). Teach: Task-driven embodied agents that chat. In Proceedings of the AAAI Conference on Artificial Intelligence (Vol. 36, No. 2, pp. 2017-2025).

Jain, U., Weihs, L., Kolve, E., Rastegari, M., Lazebnik, S., Farhadi, A., ... & Kembhavi, A. (2019). Two body problem: Collaborative visual task completion. In Proceedings of the IEEE/CVF Conference on Computer Vision and Pattern Recognition (pp. 6689-6699).

- In addition, I believe the section 2 and table 1 are biased: a majority of benchmarks mentioned here focus on learning from visual information while this paper looks at low-level textual information. The authors should acknowledge this point in the paper.

**Summary And Contributions:**

This paper looks at the problem of human-robot communication. The major contribution is a benchmark building on top of a text-based environment that evaluates the agent's ability to finish a certain task for the human given the instructions and initial human actions.

The paper look at an important problem of service robots assisting humans with their goals, and propose a new task involving goal inference based on both explicit and implicit communication. The paper is generally well written and the authors give sufficient details on the task design and dataset generation. These are the strengths of this paper. However, there are several significant weaknesses: the text-based environment overly simplified the problem, the settings are not justified and the literature review is not complete. Thus I'm reluctant to recommend acceptance.

---

> ### Author Response · Authors · 2022-08-21
> **Response to Reviewer ydPb (Part 1)**
>
> **Q1**: Too simplified, much more in the visual domain.
>
> **A1**: Please refer to our General Response Q1. We agree with the reviewer that we have simplified many recognition problems that exist in visual domains because we believe they are orthogonal to the challenges we wish to highlight in the paper. Specifically, we focus on highlighting the key challenges of goal inference and pragmatic inference in complex environments. Meanwhile, we have also discussed the potential of extending the current benchmark to visual domains in our General Response Q2.
>
> **Q2**: Motivation and application, too rare in real life.
>
> **A2**: We thank the reviewer for bringing up this discussion. We would like to first clarify that although our dataset is titled "HandMeThat," our dataset contains less-ambiguous utterances, such as "bring me the large one," and "hand me the one on the table." Please refer to Section 3.3 for more detailed discussions.
>
> Second, we would like to point out that the accuracy-brevity tradeoff (i.e., how many words do we need to use to specify the target object) exactly aligns with the algorithm we use to generate the dataset: the definition of $U_{S_k}(u, m)$ at L196 contains two terms: the first term (the utility of the target object) is corresponding to the accuracy and the second term corresponds to the cost of the utterance. In our case, the cost is exactly the number of specifiers needed to describe the object.
>
> Finally, We believe such a tradeoff exists in realistic environments. As a concrete example, assume the speaker holds a knife and stands at the counter, and there are only two apples of different sizes on the table. It's quite possible for the speaker to simply say "give me the large one" instead of "give me the large apple", since it seems clear that the speaker wants to slice something, probably an apple.
>
> We certainly agree with the reviewer that the ambiguity we construct in our benchmark represents only a limited proportion of accuracy-brevity tradeoff in real-world environments. But the experiments that the current benchmark has already highlighted important challenges for existing models. Thus, it can serve as a challenging and useful benchmark for developing learning models that can better understand human instructions in contexts.
>
> **Q3**: Why involve cost term in meaning generation?
>
> **A3**: We involve the cost term in both meaning and utterance generation, but for different reasons. The cost of utterance is used for communication efficiency. The cost of meaning, however, should be better interpreted as a regularization towards selecting a less restricted set of objects. For example, when the human wants a container for bottling fruits, potential choices for useful objects are "any jar", "any large jar", "the red jar on the table", etc. Here we assume that human prefers specifiers with fewer constraints (e.g., any jar).
>
> **Q4**: Values in fig3c do not align with equations.
>
> **A4**: Thanks for pointing out the inconsistency. There was a mistake in our definition equation -- the sign before the cost term should be negative. The correct version is $Q^*_g(s,a)=-\mathcal{C}(s,a) + V^*_g(\mathcal{T}(s,a))$, and we've revised the paper.
>
> **Q5**: Images in fig1 cause misunderstanding.
>
> **A5**: We've updated the caption of fig1 to make a clarification on this benchmark. Please refer to the revised paper.

---

> ### Author Response · Authors · 2022-08-21
> **Response to Reviewer ydPb (Part 2)**
>
> **Q6**: More explanation for section 3.3, why assume the listener knows the goal?
>
> **A6**: It's true that goal inference is an essential part of the task of our benchmark. Note that the human acts as a speaker and the robot is a listener in our episode. The speaker's goal is unknown to the listener, but in our generation process, we assume that the speaker **thinks** the listener has already inferred it correctly from world state and trajectories. To inversely solve our task, the listener could firstly do goal inference to get a $P(m|s_T, g)$, and use it as the prior distribution of the RSA listener model. We explain this part more clearly in our revised version of the paper.
>
> **Q7**: Choice of hyper-parameters.
>
> **A7**: Thanks for your suggestion. Here we explain the choice of our hyper-parameters. (Note that we've added a hyper-parameter $\alpha'$ in Section 3.3.)
>
> - The hyper-parameter $\beta_1,\beta_2,\alpha,\alpha'$ in generating meaning and utterance are chosen to balance the generated data. Intuitively, these values describe the habit of asking for help and the language use of a particular human. For example, when $\alpha'$ gets larger, the human tends to use shorter phrases in the utterance (like "hand me that"); otherwise, the human tends to use more detailed descriptions in the utterance. We choose the values so that different lengths of meaning and utterance can be generated, ensuring the diversity of data.
> - The hyper-parameter $k=10$ in the RSA model is not sensitive to our generation process. It is chosen to be non-trivial while easy to compute. This parameter is used to compute the distribution for utterance generation, and $k=10$ is already close to the effect of $k\rightarrow \infty$.
>
> **Q8**: Missing related works.
>
> **A8**: Thanks for your suggestions. We have done a more thorough literature review and included more related works. For the three references you mentioned, we compare our benchmark with theirs:
>
> - Two Body Problem[1] sets up a collaborative task where two agents can choose to communicate in both explicit (through message) and implicit (through perception) ways. Although they consider both the physical world and another agent's actions as we do, their target is finishing the task more efficiently, and only using one particular goal.
> - DialFRED[2], an extension on ALFRED, is also an instruction following benchmark, but it allows agents to actively ask questions to the human and use the response to complete tasks better. However, the command in DialFRED does not involve uncertain meanings, and the questioning is for an information-seeking procedure instead of resolving ambiguities.
> - TEACh[3] introduces a collaborative task where a *Commander* with oracle task and a *Follower* with the ability to interact with the world and communicate to complete household chores. The *Follower* needs to aggregate dialogue and action history information to decide what to do next, while the *Commander* needs to generate informative sentences as guidance. Their work extensively models the effectiveness of communication within planning tasks, but they only consider language informativeness but no communication costs.
>
> **Q9**: Not fair comparing textual tasks with visual tasks.
>
> **A9**: Thanks for the suggestion. We've revised our comparison part in the related work section and acknowledge the difference in the textual and visual interface. Although it seems unfair to juxtapose HandMeThat with other visual-based benchmarks, they are actually comparable from the perspective of language understanding and instruction following.

---

> > ### Comment · Reviewer_ydPb · 2022-08-25
> > **Response to rebuttal**
> >
> > Thank the authors for their response. Although the limitations related to the textual domain still apply, I believe the revisions have improved clarity, correctness and literature review. I'd like to increase my score by 1 point.

---

### Official Review · Reviewer_oe9r · 2022-07-23
**Interesting text-based benchmark of human instruction following and symbolic planning; room for improvement for motivation and experimental evaluation**

**Rating:** 7
**Confidence:** 4

**Strengths:**

The HandMeThat benchmark builds upon the natural distribution of human household tasks from BEHAVIOR and further enhance it with more distinct tasks (with object and object relationship randomization) and actions that change object states/relationships in the symbolic space. The latter is missing from the original BEHAVIOR work. The main differentiator of this paper seem to be the requirement to disambiguate human instructions based on physical and social contexts, which is quite unique and can be a solid contribution to the community. The difficulty level definition in Section 3.4 is also very interesting and presumpy people can use differnet partitions of the dataset for different research purposes.


**Weaknesses:**

One weakness of the benchmark could be it’s not actually physically grounded, even though it claims to be. The trajectories in the benchmark seem to be generated purely in the symbolic space.


**Additional Feedback:**

In the appendix section B: demonstrations: it’s mentioned that “we do not generate the demonstrations by PDDL planner due to the time cost of running PDDL planners on a large environment. By contrast, we directly generate a (potentially sub-optimal) solution by hand-coded policies.” Two questions: 1) since it’s an offline dataset, why is the planning time cost a concern? 2) could you give a concrete example of a sub-optimal solution? The pick-and-place example that is included seems pretty optimal.

**Clarity:**

The paper is relatively well-written, although some of the equations / concepts (e.g. action cost, cost of specifying the subgoal, etc) are difficult to understand until I also read the appendix.


**Correctness:**

The dataset/benchmark seems to be constructed in a very sound way. The supplementary material (appendix) has a very comprehensive explanation of how the dataset/benchmark was constructed.

The experimental evaluation, however, seems to be rather lacking. For the seq2seq baseline, it’s pretty unintuitive that the model performance in the partially observable scenario is better than the fully observable scenario. Although the authors attributed this to the longer text contexts in the fully observable scenario, this seems to a pure optimization issue that can be significantly alleviated with proper tuning. Also, the RL-based DRRN model completely fails even for the simplest difficulty level. The authors attribute this to the sparsity of the reward. A simple thing worth trying is to do a bit of reward shaping (since the optimal plan already exists in the dataset) and encourages the RL agent to follow the optimal action sequence. For a benchmark to be widely adopted in the community, a well-tuned set of baselines that leverage the current SOTA methods is essential for users to get started.



**Documentation:**

The documentation is through in the appendix. The benchmark code will be released under MIT license.

**Ethics:**

No ethical concerns were found

**Relation To Prior Work:**

The paper covers four areas of related work: household manipulation tasks, social reasoning, pragmatic inference, text-based reinforcement learning. For household manipulation tasks section, the paper is missing a citation of iGibson 2.0, which BEHAVIOR is built upon. For text-based reinforcement learning, it’s missing citations of many other text-based games/environments, such as LIGHT [1]. In fact, HandMeThat is more similar to Textworld and LIGHT than to BEHAVIOR or ALFRED (which have a full-blown 3D simulation environment underneath). The authors are advised to do a more through literature review of other text-based games/environments and explain why HandMeThat (in the household activities domain) should be preferred over others.

[1] Urbanek, Jack, et al. "Learning to speak and act in a fantasy text adventure game." arXiv preprint arXiv:1903.03094 (2019).


**Summary And Contributions:**

The paper introduces HandMeThat, a benchmark that evaluates human instruction understanding and following in physical and social contexts. It contains a large set of trajectories in which the human first attempt N steps to accomplish a household activity from BEHAVIOR, and then it queries the robot for help in an ambiguous way. The robot needs to resolve the ambiguity based on physical and social context and then interacts with the environment via a textual interface. The authors evalaute several baseline methods on HandMeThat and show significant room for improvement.

---

> ### Author Response · Authors · 2022-08-21
> **Response to Reviewer oe9r**
>
> **Q1**: Not actually physically grounded, purely in the symbolic space.
>
> **A1**: Thanks for pointing out the inaccurate term use. Our benchmark involves physical attributes and spatial relations, which is simulating a real-world household environment. But the object space, actions, and trajectories can all be symbolically represented. We agree that there are more physical grounding challenges in the visual domain, but we're not evaluating them in this benchmark. Please refer to General Response Q2 for more details.
>
> **Q2**: Baselines not well-tuned.
>
> **A2**: Thanks for your suggestions. We've updated extended experiments. Specifically, we tune the language-based and RL-based models to further improve baselines. For Seq2Seq model, we optimize the architecture of the observation encoder by applying a separate encoder for full observation strings, leading to much better performance for fully-observable setting. Specifically, out of 250 episodes in each split, the model can solve 56(22%), 17(7%), 10(4%), 14(6%) episodes in fully-observable setting compared with 25, 11, 3, 3 episodes originally. For partially-observable setting the partial observation strings will be fed into the new encoder, and the results also improve slightly, increasing from 53, 23, 9, 8 episodes to 64(25%), 26(10%), 10(4%), 13(5%) episodes. The partially-observable setting is still slightly better in level 1 and 2, and we attribute this to network capacity issue when the tasks are clearly specified. We have also added two extra Seq2Seq baselines given particular supervisions for reference. For DRRN model, we are training a new model with reward shaping method, and the results will be updated later. Please refer to General Response Q5 for details.
>
> **Q3**: Missing related works (text games etc.)
>
> **A3**: Thanks for your suggestion. We have updated our paper to include the suggested papers. Please refer to the revised paper.
>
> In particular, the LIGHT dataset [1] sets up a platform for studying dialogues between/among multiple agents in adventure textual-based games. It requires systems to understand actions and emotes within natural language dialogues and utterances. It is similar that they're also trying to understand the human internal idea. However, they focus on information from background knowledge such as backstory and personality, while we are considering the world state and agent's trajectory within the planning domain. Their prediction of actions and emotes can be viewed as classification into categories, while models in our task need to resolve ambiguity in instruction and then follow it.
>
> [1] Urbanek, Jack, et al. "Learning to speak and act in a fantasy text adventure game." arXiv preprint arXiv:1903.03094 (2019).
>
> **Q4**: Issues about expert demonstrations.
>
> **A4**: We agree that it is possible to use the PDDL planner to obtain the optimal plan for each task, but many goals are too complicated to be solved in a short time (in seconds). Specifically, many of the goals are composed of multiple several subgoals, and each subgoal contains nested existential or universal quantifiers. Meanwhile, our object space is also large: each scene may contain more than 200 objects. Thus, the grounding of PDDL operators and goal classifiers (which is the first step for most off-the-shelf PDDL planners) and the search will take significant time to compute some of the optimal solutions. In many cases, finding an optimal plan for a single instance takes more than 5 minutes. Thus, we have used template-based methods to obtain almost-optimal plans.
>
> Our expert demonstration can be (slightly) suboptimal when the agent needs to manipulate at least two objects. For example, suppose there is an apple and a knife on the countertop, and the goal is to put a sliced apple on the table. Our demonstration may move the apple and knife one by one to the table first and then do slicing; while the optimal solution should first slice the apple on the countertop.
>
> We agree that directly using the PDDL planning results is a better option. We will re-generate the expert demonstrations using planners in our future revisions.

---

> > ### Comment · Reviewer_oe9r · 2022-08-25
> > **Thank you**
> >
> > Thank you for your detailed response. I do not have any further questions and I'd like to keep my original rating.

---

### Official Review · Reviewer_gWGW · 2022-07-23
**Not sufficiently well motivated, too informal**

**Rating:** 5
**Confidence:** 4

**Strengths:**

- Originality:  While there are many similar datasets/benchmarks, none provide exactly the same data.
- Quality: The paper is not making any claims, so it can be considered technically sound.
- Clarity: The writing can be improved (see more details below)
- Significance: The authors are not successful in making the case for the need of such a benchmark, so I would consider the benchmark as possibly nice to have.


**Weaknesses:**

- Originality: There are many similar datasets/benchmarks. Instead of providing another benchmark, it seems to be possible to just augment the existing ones (e.g., ALFworld) with the missing text.

- Quality:
    - The paper is not making any claims. There is no theoretical justification for the work done. There are 4 level of tasks created. What are the properties of the resulting tasks? What complexity classed do they belong to?
    - The notation used through the paper is unclear and quite informal. It seems like you could mostly base your notation on the existing notation from classical planning for lifted planning tasks.
I would suggest to take a look at, e.g.,
Malte Helmert, Concise finite-domain representations for PDDL planning tasks, AIJ 2009.
If you use planning notation, you can map your resulting problems to existing planning formalisms. Once you do that, you can use existing results from planning literature on the complexity of your problems.

- Clarity: The writing should be improved (see more details below)

- Significance: You need to better justify the need for such a benchmark. Current text does not show how this benchmark helps overcoming the difficulties in communication between humans and non-human agents. The experimental results indicate that the non-trivial levels are too difficult, and it is not clear what exactly that difficulty stems from. Is it possible that you are trying to overcome too many issues at once?

**Additional Feedback:**

Questions to the authors:
1. Could you talk more about the need for this particular benchmark? How would the community benefit from it?
2. Why yet another separate benchmark? Why not, e.g., augmenting ALFworld instead?

**Clarity:**

The paper is not too clear. There is sometimes a confusion in text between human and non-human agents, since the word "agent" is used for both.
Some of the language sounds strange. For example, putting sliced fruit in a bottle? Sometimes you use "jar" and that might be a better choice. Frankly, I would just say a container.



**Correctness:**

Not relevant.


**Documentation:**

The webpage and github repo have some instructions, but I could not find any documentation for the data.


**Ethics:**

I have no ethical concerns related to this work.


**Relation To Prior Work:**

Related work section indicates the differences from prior work, as perceived by the authors. However, this is done by operating with concepts that are not clearly defined (no references either), and therefore it is not clear what is the meaning of these concepts. Social context, social reasoning, etc., what is the meaning of these concepts in this context? The citation of the book on the intentional stance is not a useful source of information in this case.
It does look like ALFRED and ALFworld could easily be augmented with the same information...


**Summary And Contributions:**

The paper proposes a benchmark and a textual environment for instruction understanding and following.
It consists of a set of "episodes", where each episode is a trace, human's goal, and two pieces of text.
The first one corresponds to the actual meaning of the human request from the agent and the second one is the text of the ask given to the agent.

---

> ### Author Response · Authors · 2022-08-21
> **Response to Reviewer gWGW (Part 1)**
>
> **Q1**: Why yet another separate benchmark, not augmenting ALFWorld?
>
> **A1**: We did not choose to augment the ALFWorld dataset for two main reasons: the limited number of goal templates, and the lack of realistic task distributions. First, ALFWorld contains only 7 templates and they are relatively short-horizon compared to BEHAVIOR tasks (e.g., most templates manipulate only one single object: heat X, cool X, put X into Y, etc.) This setting makes it hard to generate partial human trajectories and ask robots to help. Second, tasks in ALFWorld are generated by substituting slots in task templates with all possible object categories. As a result, it is usually impossible to infer underspecified goals from historical actions: consider the task put a heated apple in X. In our setting, following the BEHAVIOR task distribution, X needs to be a container for food and large enough to hold the apple. However, there are no such constraints in ALFWorld (i.e., X can be kitchen sinks or trash bins). However, we have tried hard to keep the interface of our environment similar to ALFWorld so that most algorithms should be applicable.
>
> **Q2**: Notation using and complexity of tasks.
>
> **A2**: Thanks for your suggestion. We have updated part of our notations using notations from classical planning literature. We introduce the concept of state variables to better define our state and action representation. We didn't use the same definition of PDDL operators for our actions because our object-centric representation can better describe the later non-planning parts. We provide the formal definitions for predicates and actions in our [domain file](https://github.com/Simon-Wan/HandMeThat-Release/blob/master/handmethat/envs/robot_domain.pddl).
> Many other notations are out of PDDL planning domain, e.g. generating human meaning and utterance. Besides, all the goals in our task are in first-order logic. Rather than using the free variables notations, we provide goal templates with additional notation like **?[fruit]** to make the presentation more concise. That's because all these variables will be grounded at the initial sampling stage, and the goal for agents in our episode is a certain instance of some goal template.
>
> As for the complexity issue, the episodes in HandMeThat are not pure planning tasks. Instead, the key challenge of HandMeThat is "**learning**" to resolve language ambiguities based on training data. The four hardness levels we define in the paper disentangle the challenges into 1) textual-based planning, 2) goal inference, 3) pragmatic language reasoning, representing different parts of abilities to be **learned** by models. If we assume that object states and relations can be extracted from texts as symbolic representations and the goal is fully-specified without ambiguity, the remaining problem is reduced to a classical PDDL planning problem, belonging to PSPACE-complete complexity class. Actually, our expert demonstration is generated by PDDL planning algorithms (more specifically, A-Star search with hFF heuristics).
>
> **Q3**: Clarity issue.
>
> **A3**: Thanks for pointing out the confusing words in our paper. We have changed our paper by avoiding using the term "agent" when it's unclear. We have also made some minor changes to other language use, and please refer to the revised paper for details.

---

> > ### Comment · Reviewer_gWGW · 2022-08-22
> > **Complexity**
> >
> > It is true that planning is PSPACE-complete in general even for simplest STRIPS fragment. But do you use the full fragment? Or maybe your tasks belong to more restricted fragments? For instance, STRIPS without delete effects is NP-complete for bounded planning and in P for plan generation (when cost/length of plans is not considered).
> >
> > Also, please avoid using search algorithms for cost-optimal planning (such as A*) with inadmissible heuristics, such as FF. The result is a slow greedy (non-optimal) planner, that does not provide any guarantees on plan quality. It's akin to taking the worst of two worlds - the slowness of cost-optimal planning and the lack of guarantees on solution quality of satisficing planning.
> > If you need optimality, please use A* with admissible heuristics (e.g., LM-cut), if you do not require provably optimal solutions, I suggest using greedy best-first search (GBFS) instead of A*.

---

> > > ### Author Response · Authors · 2022-08-24
> > > **Response to Reviewer's Comment -- Complexity Issue**
> > >
> > > Thank you for pointing these out!
> > >
> > > 1. "Unfortunately", the operators in HandMeThat do have delete effects so we cannot apply this given complexity results. We are also unaware of any better complexity results that can be applied to our task.
> > >
> > > 2. We agree that generating the expert demonstrations with hFF cannot guarantee optimality. We are planning to use A* with LM-Cut to generate optimal demonstrations in our official release.

---

> > ### Comment · Reviewer_gWGW · 2022-08-22
> > **Augmenting ALFWorld?**
> >
> > Are you saying that you couldn't extend ALFWorld with additional goal templates and task distributions?
> > If you could, I imagine that would be helpful beyond your focus on understanding human intentions.

---

> > > ### Author Response · Authors · 2022-08-24
> > > **Response to Reviewer's Comment -- Why not augmenting ALFWorld**
> > >
> > > ALFWorld could not be extended with the goal templates in HandMeThat/BEHAVIOR because there are object categories (e.g., rag), physical states (e.g., soaked, sliced), and action types (e.g., clean {obj} with {tool}, slice {obj} with {tool}) that do not exist in its vision and physics simulator (AI2Thor).
> > >
> > > On the other hand, if we only consider the text-based environment, our environmental interface is identical to theirs, except that our initial observation contains additional human action history. In fact, our Seq2Seq model uses the same architecture as the one in the ALFWorld paper.

---

> ### Author Response · Authors · 2022-08-21
> **Response to Reviewer gWGW (Part 2)**
>
> **Q4**: Better justify the need for such a benchmark.
>
> **A4**: The existing benchmarks cannot evaluate the abilities we are focusing on---resolving ambiguous instructions based on both the physical world state and actions and goals of another agent (human). Developing a benchmark for such scenarios is challenging in both data collection and automatic evaluation. Specifically, it’s hard to collect the everyday dialogues with corresponding physical states; and it is hard to build automatic evaluation protocols that involve human judgments about robot success. HandMeThat is a first step toward building such a benchmark: it leverages the availability of human-annotated real-world task distributions (BEHAVIOR) and computational models for human language modeling (RSA models) and creates a benchmark for contextual language understanding with physical and social information (i.e., actions and goals of humans) that supports automatic data generation and evaluation.
>
> Another important feature of the proposed benchmark is that it supports fine-grained "diagnostics" of models. HandMeThat disentangles different challenges by splitting the dataset into four hardness levels. The challenges include planning, goal inference, and pragmatic language reasoning. The data split here can be separately used for studying different modules.
>
> Finally, since the episodes in our benchmark can be symbolically represented, it can be potentially rendered into different interfaces, such as images, too. Therefore, it's possible to extend other challenges within the visual domain to our pipeline in future works.
>
> **Q5**: Experiments didn't show what exactly that difficulty stems from.
>
> **A5**: Intuitively, one of the most challenging parts for models in HandMeThat tasks is learning to predict human intention. As shown in General Response Q5, we have added two extra baselines given particular supervisions---either the goal or the subgoal of human is explicitly provided along with the utterance, without any special design in encoding them. Specifically, the success rate of both fully and partially observable models (in level 2 and above) improve from around 5% to around 15% only by appending the subgoal in first-order-logic format to the inputs. Also, the difference in performance among these three levels gets smaller. The results highlight that inferring human goals and subgoals is helpful in resolving ambiguous instructions.
>
> On the other hand, the tasks in HandMeThat naturally exist in real life and our human test shows that they can be decently accomplished by a human. Our benchmark aggregates the challenges in planning, goal inference, and pragmatic language reasoning. On the other hand, these challenges are disentangled by splitting the benchmark into four hardness levels. Since we have plenty of supervision (e.g., goal, subgoal, expert demonstration) for each episode of data, it is possible to locate the challenges within the pipeline intuitively when we are designing better models.
>
> **Q6**: Unclear concepts in the related work section.
>
> **A6**: Thanks for your suggestions. We've revised the use of some confusing concepts. Specifically, we avoid using the term "social contexts" due to ambiguity, and we clarify the meaning of "social reasoning" in our domain. Please refer to our revised paper.
>
> **Q7**: Data documentation.
>
> **A7**: Thanks for your suggestion on data documentation. The details of the HandMeThat task setting are presented in supplementary materials. We will also update our GitHub page to include the JSON structure of data.

---

> > ### Comment · Reviewer_gWGW · 2022-08-22
> > **Justify the need, not explain how your benchmark is different from what's there**
> >
> > To justify the need for your benchmark, you need to claim (and show) that your benchmark helps solving an *important* problem.
> > Current text does not show how this benchmark helps overcoming the difficulties in communication between humans and non-human agents.

---

> > > ### Author Response · Authors · 2022-08-24
> > > **Response to Reviewer's Comment -- Justify the need of this benchmark**
> > >
> > > Thank you for your comments. HandMeThat is a benchmark for a **realistic** and important problem in human-robot communication. In real-world communication, ambiguous instructions are usually used. For example, David Chapman documented in his book [1] that human in video-game playing uses utterances such as "don't do that," "go get that item," "go the other way," etc. Understanding these instructions is an important prerequisite for robots that can collaborate with humans flexibly, e.g., in household environments, and this is exactly the problem we want to highlight with our environment.
> > >
> > > HandMeThat is a suitable environment to study this problem for the following three reasons. First of all, evaluation of robot performance in human-robot communication and collaboration is hard, because that generally requires humans in the loop. HandMeThat is the first diagnostic dataset of its kind that supports the automatic generation of ambiguous instructions and automatic evaluation of robot performance. Second, it allows researchers to isolate the problem of ambiguous instruction understanding by bypassing certain confounding difficulties, such as visual perception, state estimation, etc. It also supports a "symbolic interface" if researchers want to bypass the language understanding problem. Finally, its difficulty levels allow researchers to focus on different aspects of the problem: level 1 focuses on language understanding and planning alone; level 2 adds in human goal recognition; level 3 adds in pragmatic reasoning; level 4 requires robots to ask additional questions back to gather information.

---

> > ### Comment · Reviewer_gWGW · 2022-08-22
> > **What the difficulty stems from**
> >
> > What you call goal inference is called goal recognition in planning and there is ample literature on the topic. One possible approach is to compile goal recognition into a planning problem (you can read about it here https://www.ijcai.org/proceedings/2021/0616.pdf), so these are basically different sides of the same coin.
> > So, basically, your task splits into planning and natural language understanding.
> > It might be the case that this is difficult only as a unified task and can be made simpler by splitting. In other words, there might be the case that putting these tasks into this benchmark set you create an unnecessary difficulty. I am not saying that's the case, but it is not clear that it isn't.

---

> > > ### Author Response · Authors · 2022-08-24
> > > **Response to Reviewer's Comment -- What the difficulty stems from**
> > >
> > > Thanks for your feedback!
> > >
> > > We have been using the term "goal inference" following "Goal Inference as Inverse Planning" (CogSci 2007) and "Online Bayesian Goal Inference for Boundedly-Rational Planning Agents" (NeurIPS 2020). However, we agree that the term “goal recognition” is more popular in literature. We will revise the paper accordingly.
> > >
> > > We also agree with you that our task can be viewed as a combined task of natural language understanding (NLU) and planning. However, this is an important task that can not be **decomposed** into NLU and planning and then be solved independently.
> > >
> > > * From the language understanding perspective, while many related papers have been studying language grounding in objects and actions, our paper especially focuses on language understanding which involves understanding the internal goals of humans.
> > >
> > > * From the planning/goal recognition perspective, language inputs are informative in goal recognition. For example, the type of tool that the human asks you to hand over should be informative for your recognition of her goal.
> > >
> > > We are not exactly sure about what you meant by "can be made simpler by splitting." If you are suggesting that a system can first translate language inputs into symbolic representations (e.g., from human instructions to first-order-logic subgoal specifications), and then solve the planning/goal recognition problem separately, there are four reasons why this is not the ideal setup.
> > >
> > > 1. It is true that all language instructions in our dataset has an underlying first-order-logic (FOL) formula. However, semantic-parsing natural language into FOL formula may not be practical in real-world scenarios. Thus, exploring how we can use latent representations for language (e.g., neural networks) is an important direction.
> > > 2. More importantly, even if one has parsed instructions into FOL, this does not really simplify the problem. Specifically, the formula will still be an under-specified subgoal (e.g., "hand me that"). Thus, the goal recognition algorithm still needs to consider both human historical actions and the subgoal formula.
> > > 3. Furthermore, the goal recognition algorithm still needs to learn commonsense knowledge about what are possible goals. That is, models should not assume access to the list of all possible goals. This is an important and realistic setting because, we can never manually program all possible tasks that a human could ever do, in a real-world robot factory. Thus, developing robots that can learn such commonsense from experience is important.
> > > 4. So far we have separated the goal recognition problem and the planning of robot actions. But this is not ideal either. Because, in partially-observable environments, the robot can and should take actions to gather additional information to facilitate his goal recognition.
> > >
> > > Overall, understanding ambiguous instructions is a real-world problem for human-robot communication, and there naturally exists interplays between different modules. Thus, this problem should be studied as a whole. However, we do strongly agree with you that investigating how off-the-shelf algorithms for language understanding and goal recognition perform in our environment is an important research question. We did not try to directly apply off-the-shelf goal recognition algorithms (e.g., methods listed in the goal recognition survey paper) because they do not scale up to our goal space (>300,000).

---

> > > > ### Comment · Reviewer_gWGW · 2022-08-25
> > > > **So it seems to be about goal recognition from text**
> > > >
> > > > I think we have finally distilled what the core problem you are trying to tackle here: goal recognition from (ambiguous) textual instructions. It is possible to separate the planning problem from the goal recognition, assuming partial knowledge about the recognized goals. Then the planning agent has to resolve the ambiguity during planning phase.
> > > > I would strongly encourage you to formally specify the problem(s) and position it with respect to the research done on goal recognition. Being formal here will help, I am sure, to tackle the actual difficulty and meaningfully measure the progress towards overcoming that difficulty.
> > > >
> > > > Thank you for your replies and the explanations. I am certain that integrating these clarifications into the paper will significantly improve it. Currently, I still feel that the paper is not ready for publication in its current form.

---

> > > > > ### Author Response · Authors · 2022-08-28
> > > > > **Thanks for your constructive suggestions! Manuscript updated.**
> > > > >
> > > > > Dear Reviewer gWGW,
> > > > >
> > > > > Thanks for your consideration and constructive comments, which have helped us greatly improve the quality and clarity of our paper. We agree that making more connections to goal recognition literature would help set up the background of our work. Therefore, based on our earlier discussions, we have updated our paper with the following revisions. Newly updated texts are highlighted in blue.
> > > > >
> > > > > 1. In the introduction (Section 1), we added a new paragraph formalizing the problem as "recognition of human's subgoal from her historical actions and ambiguous instructions."  Furthermore, that paragraph highlights how different "subproblems" in our tasks (goal recognition, pragmatic reasoning, and planning) interplay with each other.
> > > > > 2. We added a new paragraph in Section 2 with more references to goal recognition literature. We also discuss the main difference between our benchmark and related work. Specifically, the inputs to our robot contain both human actions and ambiguous instructions. And our objective is not to fully recover the internal goal of the human, but only the subgoal set up by the human.
> > > > > 3. We reformat and formalize the notations for the domain, goal, and human trajectory (including both the introductory paragraphs in Section 3 and Section 3.1) using the existing notations in the classical planning literature.
> > > > > 4. We adopted a more formal definition of the "subgoal" concept (Section 3.2) based on first-order logic formula, and mathematically define the "usefulness" of a subgoal w.r.t. a goal.
> > > > > 5. We added a new subsection (Section 3.5) to describe the important challenges of our benchmark, by summarizing our response to your questions previously.
> > > > >
> > > > > We hope that these clarifications and formalizations would better clarify the problem setups and highlight the challenges of HandMeThat. We appreciate your helpful suggestions. Please don’t hesitate to let us know if you have any additional questions or comments!

---

> > > > > > ### Comment · Reviewer_gWGW · 2022-08-28
> > > > > > **Your changes**
> > > > > >
> > > > > > Two more comments:
> > > > > > 1. "Another important difference between our work and other goal recognition tasks is that our robots can not access the list of all subgoals a priori. Instead, they need to learn from data the commonsense knowledge of plausible human goals. Building agents that can learn to recognize novel goals is an important desideratum towards household robots that can adapt to novel environments and tasks"
> > > > > > While I realize that such statements are very common in learning community and rarely challenged, this restriction sounds very artificial to me. This is not the main obstacle for adapting to novel environments. If the information is crucial for building the environment, there is no reason not to expose it to the agent. Not exposing the information just for the sake of artificially making the robots job harder does not sound like a good practice to me.
> > > > > > 2. It sounds a bit strange to me to use "he/him/his" as robot's pronouns. is that the common practice? I honestly don't know.
> > > > > >
> > > > > > Regardless of the comments, in light of your changes, I have decided to raise my score.

---

> > > > > > > ### Author Response · Authors · 2022-08-28
> > > > > > > **Clarification on Learning**
> > > > > > >
> > > > > > > Thanks for the comments. We fully agree that this is an important point and wish to clarify.
> > > > > > >
> > > > > > > As an evaluation benchmark for real-world human-AI communication, we emphasize the importance of learning human's underlying goal space. Indeed, much of the information (e.g. planning domain and transition rules) for building the environment can be programmed "in the robot factory." However, we believe that adapting to the novel goal of space is still a crucial and inevitable challenge for robots in the real world.
> > > > > > >
> > > > > > > In a household environment, deploying a robot requires the adaption to the habits of new family members. For example, consider the context that a human just finished cooking noodles, and she asks the robot "can you get me a plate and utensils?" The type of utensil to be taken varies from person to person---an Italian family may prefer forks, while a Japanese family may need chopsticks.
> > > > > > >
> > > > > > > We strongly agree that it is important to investigate how we can pre-program some of the goal-space knowledge (e.g., basic types of objects and their common usage, universal task categories, etc.) into models, while still allowing robots to update their knowledge in real world. We will definitely incorporate these clarifications into our main paper.
> > > > > > >
> > > > > > >
> > > > > > > As for the pronoun choice, we do not think there is a convention for this. In this paper, we use she/her for the human and he/his for the robot because they are two symmetric parties in communication. There are some other choices, too. For example, in WatchAndHelp, the authors name their agents Alice and Bob, in CerealBar and TEACh, the authors used gender-neutral pronouns (they/their) for the robots.

---

> > > > > > > > ### Comment · Reviewer_gWGW · 2022-08-29
> > > > > > > > **Not convinced at all**
> > > > > > > >
> > > > > > > > Even if you expect to observe new objects not occurring in your training, the predicates do not change, and therefore once you have identified the new objects, the space of possible goals is identified as well.

---

> > > > > > > > > ### Author Response · Authors · 2022-08-29
> > > > > > > > > **Not about new objects but goals**
> > > > > > > > >
> > > > > > > > > Thanks for your time and consideration. We clarify that the point here is not about novel objects. Even if we know all objects categories, there will always be differences between "possible goals" in different household environments. For example, the utensils that different families use to eat noodles, how different families set up their dining tables, how they pack camping bags, and how they organize decorations in rooms, may differ. A household robot should learn such knowledge to best help humans. Therefore, we think it is important to build models that can adapt to novel sets of possible goals from experience.

---

> > > > > > > > > > ### Comment · Reviewer_gWGW · 2022-08-29
> > > > > > > > > > **Not quite**
> > > > > > > > > >
> > > > > > > > > > While the goals are "new", they are still logical formula over ground predicates (grounded in the new objects), so unless you deliberately hide an information about the predicates, once you have a set of objects and your restrictions on the logical formula considered, you have your set of possible goals.
> > > > > > > > > > In most scenarios, the goals are just conjunctions of ground predicates, and while it is possible to encode more complex formula, it leads to complex environments that RL agents will not be able to handle anyway. RL community starts to realize it as well, see for example the recent IJCAI 2022 paper https://www.ijcai.org/proceedings/2022/0507.pdf

---

> > > > > > > > > > > ### Author Response · Authors · 2022-08-29
> > > > > > > > > > > **Goals are not new, but human preferences are**
> > > > > > > > > > >
> > > > > > > > > > > Thank you for your response and the pointers. We agree that domain information such as object categories, predicates, and actions can be defined in robot factories. Thus, it is true that the set of ALL possible goals can be generated by logic formulas. However, in reality, a human's internal goal is not sampled uniformly from ALL possible goals that can be written in a logic language. Instead, they follow certain personalized distributions.
> > > > > > > > > > >
> > > > > > > > > > > For example, consider the following two goals:
> > > > > > > > > > >
> > > > > > > > > > > - $\exists x.\exists y. \text{noodle}(x)\land \text{fork}(y)\land \text{next-to}(x, y)$ (prepare noodles and put a fork on the side)
> > > > > > > > > > > - $\exists x.\exists y. \text{noodle}(x)\land \text{chopsticks}(y)\land \text{next-to}(x, y)$ (prepare noodles and put a pair of chopsticks on the side)
> > > > > > > > > > >
> > > > > > > > > > > Consider you are a household robot and just got deployed at a new family house. After Alice (the human) cooks some noodles and asks you "can you bring me some utensils," you need to make a decision about which utensils to pick up (forks, chopsticks, etc.) If you are new to the house, you probably have a uniform distribution over two options (fork and chopsticks). However, such prior over humans' internal goals should be updated over time based on the robot's interactions with humans.
> > > > > > > > > > >
> > > > > > > > > > > The same is true for many other tasks. It is important for household robots to learn how different families organize dining tables, such that when they hear the instruction "put the forks onto the dining table," they should understand where they should put the forks (e.g., to the left or to the right of the plates, depending on the human's dominant hand).

---

> > > > > > > > > > > > ### Comment · Reviewer_gWGW · 2022-08-29
> > > > > > > > > > > > **Very similar to goal recognition in planning**
> > > > > > > > > > > >
> > > > > > > > > > > > You probably want to add  \land \text{at}(x, dining-table)\land \text{at}(y, dining-table) to these goals... You don't want to end up with the food prepared nicely in the kitchen sink (or worse). These goal facts do not appear in the textual description but should definitely be recognized. This all is handled in goal recognition in various settings.
> > > > > > > > > > > >
> > > > > > > > > > > > It feels more and more to me like you are trying distance yourself from the existing work on goal recognition in planning. If you do that because you worry that your work is not innovative enough in the presence of that body of work, I would not worry about that. Instead, I would embrace the existing research and think about ways to exploit it better in this framework.

---

> > > > > > > > > > > > > ### Author Response · Authors · 2022-08-29
> > > > > > > > > > > > > **Goal recognition with learned prior distribution**
> > > > > > > > > > > > >
> > > > > > > > > > > > > Dear Reviewer,
> > > > > > > > > > > > >
> > > > > > > > > > > > > Thank you for your response. We are not trying to differentiate ourselves from goal recognition. Instead, we are trying to state some potential difference and suggest the integration of planning algorithms with learning methods. To make our discussion a bit more concrete, let's consider the probabilistic formulation of goal recognition as in [Ram´ırez and Geffner, 2010] (which is also summarized in Section 6.1 of the survey paper you linked [Meneguzzi and Pereira, 2021]).
> > > > > > > > > > > > >
> > > > > > > > > > > > > In [Ram´ırez and Geffner, 2010], a goal recognition algorithm outputs a probability distribution $p(G|\Omega)$, where $G \in \mathcal{G}$ are possible goals, and $\Omega$ is robot's observation (e.g., human trajectories). To do that, they compute $p(G|\Omega) \propto p(G) p(\Omega | G)$. Here, the term $p(G)$ corresponds to robot's prior distribution over "all possible goals that a human can perform." For simplicity, let's say the goal space $\mathcal{G}$ here contains all possible first-order-logic formulas composed of objects and domain predicates.
> > > > > > > > > > > > >
> > > > > > > > > > > > > It is true that many prior distributions can be programmed into the robot in the robot factory (e.g., well-cooked food should not be in kitchen sinks), but there is still much other prior information that should be learned in the wild. For example, as discussed in our previous response, whether the person (Alice) prefers using forks or chopsticks. Such prior vary from person to person, and can not be pre-programmed in a robot factory.
> > > > > > > > > > > > >
> > > > > > > > > > > > > This already suggests an integrated solution of goal recognition algorithms (e.g., [Ram´ırez and Geffner, 2010]) and learning algorithms for our problem setting. That is, instead of doing end-to-end learning to recognize the subgoals specified by the human, one could consider only learning priors over goals/subgoals and combine such prior with existing goal recognition algorithms (after bypassing language understanding challenges such as RSA reasoning).
> > > > > > > > > > > > >
> > > > > > > > > > > > > Based on the reading of the suggested survey and some of our searches, we did not find a good discussion in this field. If you have any pointers that you can share, please kindly let us know and we should definitely add them to the paper.
> > > > > > > > > > > > >
> > > > > > > > > > > > > [Ram´ırez and Geffner, 2010] Miquel Ram´ırez and Hector Geffner. Probabilistic Plan Recognition Using Off-the-shelf Classical Planners. In AAAI, 2010.
> > > > > > > > > > > > >
> > > > > > > > > > > > > [Meneguzzi and Pereira, 2021] Felipe Meneguzzi and and Ramon Fraga Pereira. A Survey on Goal Recognition as Planning. In IJCAI, 2021.

---

> > > > > > > > > > > > > > ### Comment · Reviewer_gWGW · 2022-08-29
> > > > > > > > > > > > > > **That was exactly my point**
> > > > > > > > > > > > > >
> > > > > > > > > > > > > > I think we are converging to the same point.
> > > > > > > > > > > > > >
> > > > > > > > > > > > > > One additional reference I might suggest is IJCAI 2016 paper "Plan recognition as planning revisited" by Sohrabi et al., that extends over Ramirez and Geffner AAAI 2010.
> > > > > > > > > > > > > >
> > > > > > > > > > > > > > I am not completely sure what discussion are you referring to. Is it about combining externally learned priors and the probability distribution over goals? I think that might be handled by specific cost functions in the transformed planning task (when solving goal recognition as planning). There might be some discussion about that in the paper by Sohrabi et al, but for a different (albeit very similar) purpose of accounting for noisy observations.

---

> > > > > > > > > > > > > > > ### Author Response · Authors · 2022-08-29
> > > > > > > > > > > > > > > **Thanks for the pointer! Manuscript updated**
> > > > > > > > > > > > > > >
> > > > > > > > > > > > > > > Dear Reviewer gWGW,
> > > > > > > > > > > > > > >
> > > > > > > > > > > > > > > We are glad that we reached the same point! We have updated the discussion in Section 3.5. Specifically, we have rephrased the challenge as suggesting learning human preferences from experiences and discussing the promising direction of integrating external learning algorithms with existing goal recognition work (Sohrabi et al.). It is definitely interesting to see how prior probability distributions over goals can be integrated in a similar way into the planning domain.
> > > > > > > > > > > > > > >
> > > > > > > > > > > > > > > Thanks again for your time and the fruitful discussion.
> > > > > > > > > > > > > > >
> > > > > > > > > > > > > > > Authors

---

### Official Review · Reviewer_Hz1L · 2022-07-26
**Bechmark for HRI ambuguity**

**Rating:** 6
**Confidence:** 3
**Correctness:** Yes
**Clarity:** Generally clear

**Strengths:**

1. It’s very important for robots to understand human instructions under uncertainty.
2. Paper is generally well-written for readers to understand the main contributions.


**Weaknesses:**

1. The proposed setting is too simplified for the real HRI use-cases. Humans use ambiguous language instructions, often accommodating those with non-verbal communication. For example, pointing/eye gazing.
2. The proposed benchmark operates in a text game. This setting is analogous to state-space,  which does not have any ambiguities. However, this benchmark aims to resolve ambiguities in human instructions. It seems this setting kind of defeats its own purpose.
3. The proposed ambiguities in the benchmark are quite narrow and limited. They are essentially hand-crated by the author. I am unsure if they truly represent the ambiguities in real HRI settings. The only justification authors present is that these templates are from behavior challenges. However, behavior challenges also have limitations (discrete states etc)



Resolving ambiguities in human instructions is very important for the HRI community.
However, there is a lack of reasonable justification on how this proposed text-game setting will benefit HRI.  From my perspective, real-world HRI has much more than textual settings.



**Additional Feedback:**

N/A

**Documentation:**

N/A

**Relation To Prior Work:**

Yes

**Summary And Contributions:**

This paper proposed a new dataset and a text game to resolve instruction ambiguities for indoor task planning in HRI setting.  The robot is supposed to understand human utterances and propose proper actions to help humans. The tasks are from behavior challenges. However, all actions are operating in the state space through a textual game setting.

---

> ### Author Response · Authors · 2022-08-21
> **Response to Reviewer Hz1L**
>
> **Q1**: Other info besides texts.
>
> **A1**: Thanks for your suggestion. We agree that there are many additional, non-verbal channels for communication. In the benchmark we aim at taking the first step: we particularly focus on verbal communication with ambiguities. The advantage of HandMeThat is that it is a benchmark that supports automated evaluation, and is fully controllable and annotated with ground-truth information such as agents' internal goals and object states. It is not intended for directly training agents that operate in the real world, but for an automatic evaluation platform. Please also refer to General Response Q1 for details.
>
> **Q2**: Why resolve ambiguities in an unambiguous state-space environment?
>
> **A2**: First, the proposed benchmark has a partially-observable setting, where the agent initially has uncertainty about the world state. For example, agents can not see through closed containers. In this case, the agent needs to explore the environment (through text commands) and find relevant objects.
>
> Second, we believe that ambiguities in language specification and uncertainty in states (e.g., object segmentation and property prediction) are two different aspects of the problem, and can be potentially studied separately. Specifically, in this paper, we have chosen to minimize the difficulty of recognizing object states from texts and focused on learning to infer human goals and resolve ambiguous utterances.
>
> Finally, it is also possible to extend the current framework to continuous and partially-observable inputs such as images. In this way, the orthogonal aspects such as visual recognition and non-verbal communication can be involved. Please see our general response Q2 for discussions.
>
> **Q3**: Hand-crafted, not truly represent the real world (even with BEHAVIOR).
>
> **A3**: We would like to clarify that HandMeThat is not designed for exactly model real-world human-robot interaction. Instead, HandMeThat should be used as a diagnostic benchmark for developing models and inspecting different aspects of models. Thus we have been focusing on a systematical evaluation of models that can interpret ambiguous language instructions considering both the physical contexts and actions and goals of humans. Our benchmark leverages the availability of human-annotated real-world task distributions from BEHAVIOR, which entails the ability of understanding common-sense prior knowledge. This is an essential component of social reasoning and can be evaluated by our benchmark. Please refer to General Response Q1 for more details.
>
> **Q4**: How does real-world HRI benefit from it?
>
> **A4**: We rewrite part of our paper to better justify our motivation as well as how the community can benefit from this benchmark. The existing benchmarks cannot evaluate the abilities we are focusing on---resolving ambiguous instructions based on both the physical world state and actions and goals of another agent (human). Developing a benchmark for such scenarios is challenging in both data collection and automatic evaluation. Specifically, it’s hard to collect the everyday dialogues with corresponding physical states; and it is hard to build automatic evaluation protocols that involve human judgments about robot success. HandMeThat is a first step toward building such a benchmark: it leverages the availability of human-annotated real-world task distributions (BEHAVIOR) and computational models for human language modeling (RSA models) and creates a benchmark for contextual language understanding with physical and social information (i.e., actions and goals of humans) that supports automatic data generation and evaluation.
>
> Another important feature of the proposed benchmark is that it supports fine-grained "diagnostics" of models. HandMeThat disentangles different challenges by splitting the dataset into four hardness levels. The challenges include planning, goal inference, and pragmatic language reasoning. The data split here can be separately used for studying different modules.
>
> Finally, since the episodes in our benchmark can be symbolically represented, it can be potentially rendered into different interfaces, such as images, too. Therefore, it's possible to extend other challenges within the visual domain to our pipeline in future works.

---

> > ### Author Response · Authors · 2022-08-26
> > **Looking Forward to Your Feedback**
> >
> > Dear Reviewer,
> >
> > Thank you again for the constructive reviews, which have helped us greatly improve the quality and clarity of our paper. We hope our response has been able to address your concerns. We have also updated our manuscript accordingly. As we approach the end of the discussion period, please don’t hesitate to let us know if you have any additional questions or comments!
> >
> > Thanks for your time,
> >
> > Authors

---

> > > ### Comment · Reviewer_Hz1L · 2022-08-29
> > > **feedback for HandmeThat**
> > >
> > > Thanks for the detailed response in various different sections. The rebuttal partially addressed some of the concerns, even though the main concern still exists(a huge amount of simplifications). I feel this benchmark could be of some use to the community.
> > >
> > > On the other hand, I do not believe it's easy to extend the current benchmark to the vision or robotics domain. Generating reasonable trajectories in complex indoor scenes with complex object manipulations(w or w/o abstract grasping) is non-trivial.
> > >
> > > Therefore, I increase my rating by 1.

---

### Official Review · Reviewer_eyB2 · 2022-07-28
**An interesting problem**

**Rating:** 6
**Confidence:** 3
**Correctness:** Seems correct.

**Strengths:**

1. The HandMeThat problem is interesting and well formulated. I think it can provide a useful benchmark platform for comparing RL based models in a (limited) scenario.
2. Benchmark results using RL methods and other non-learning based alternatives are provided.


**Weaknesses:**

One limitation about this paper is that human tests are missing during the comparison experiments. I assume this would be difficult to collect data from humans, but it would be nice to see if humans are able to do this at all. Within a limited time, it seems like even humans might find those tasks too difficult. I feel that learning based methods also did not prove the potential under uncertainty in the reported results.

**Additional Feedback:**

Nothing particular.


**Clarity:**

Fairly clear.


**Documentation:**

The link to the website is provided.


**Ethics:**

Not applicable.


**Relation To Prior Work:**

The paper discussed and compared with the prior work sufficiently.


**Summary And Contributions:**

This paper proposes a new robot learning environment for instruction understanding and following within physical and social contexts. Compared with current existing methods that focus on the language grounding, the paper added instruction understanding and following. The problem is formulated as a Markov decision process in the textual world. Benchmark results use a random agent, a heuristic-based agent, a sequence-to-sequence model, and  a DRRN RL model. The reported results indicate that the new robot learning environment with consideration to instruction understanding and following in physical and social contexts is challenging and the heuristic model showed the best average performance in the fully-observable setting while the sequence-to sequence showed the best average performance in the partially-observable setting.

---

> ### Author Response · Authors · 2022-08-21
> **Response to Reviewer eyB2**
>
> **Q1**: Human performance.
>
> **A1**: Thanks for your suggestion. It’s surely hard to collect the everyday dialogues with corresponding physical states directly. Alternatively, we choose to it leverage the availability of human-annotated real-world task distribution from BEHAVIOR.
> We've also evaluated the human performance on our benchmark. The results show that our human subjects can perform well on tasks in level 1 and 2, showing a success rate of more than 80% in fully-observable setting. In level 3, the inference of speaker intention gets more difficult. As a result, the performance drops significantly to 36% in fully-observable setting, but still outperforms our learning baselines. Tasks in level 4 are intrinsically under-specified. Thus, the human subjects perform worse than level 3. Success rates in partially-observable setting are also significantly lower than in fully-observable setting, suggesting that observability is also a critical challenge. Note that our environment supports robots asking additional clarification questions to the simulated human in order to resolve ambiguities. However, since no current learning models are not capable of actively asking questions, for a fair comparison, we do not enable this option in human experiments.
> Generally speaking, the human performances suggest that the generation process is reasonable, and the data splits present the hardness differences. Please refer to General Response Q4 for more details.
>
> **Q2**: Learning-based methods did not prove the potential.
>
> **A2**: Thank you for the suggestions on tuning baselines! We have updated extended experiments. Specifically, we tune the language-based and RL-based models to further improve baselines. We change the observation encoder of our Seq2Seq model and the performances improve significantly, especially for fully-observable setting. Specifically, out of 250 episodes in each split, the model can solve 56(22%), 17(7%), 10(4%), 14(6%) episodes in fully-observable setting compared with 25, 11, 3, 3 episodes earlier. The partially-observable results also improve from 53, 23, 9, 8 episodes to 64(25%), 26(10%), 10(4%), 13(5%) episodes. However, we believe that a better design of modules for encoding physical and social information is needed, and there is still significant room for improvements. Please refer to General Response Q5 for more details.

---

> > ### Author Response · Authors · 2022-08-26
> > **Looking Forward to Your Feedback**
> >
> > Dear Reviewer,
> >
> > Thank you again for the constructive reviews, which have helped us greatly improve the quality and clarity of our paper. In our revision, we have included new results on the human performance of our task. We have also updated a few baseline results based upon your suggestions on baseline tuning. We hope our response and new results have been able to address your concerns. As we approach the end of the discussion period, please don’t hesitate to let us know if you have any additional questions or comments!
> >
> > Thanks for your time,
> >
> > Authors

---

### Official Review · Reviewer_UhYr · 2022-07-28
**A novel text-based instruction following benchmark. The creation process of this benchmark is very interesting.**

**Rating:** 7
**Confidence:** 3
**Correctness:** The data set is constructed in a soun…
**Clarity:** Writing is clear with plenty of detai…

**Strengths:**

- The proposed benchmark focuses on very critical topics in human-robot interaction: the trade-off between informativity and communication cost in human language, and the challenges in understanding human intention and resolving ambiguities.
- This paper presents a systematic way to create and control the level of ambiguity in instruction following tasks. And this approach seems can be easily generalized to wider applications.
- This paper provides great details and explanations about the creation process of the benchmark, which are beneficial to researchers in this field.

**Weaknesses:**

- While this paper presents an interesting benchmark, but the experiments are not so informative. The heuristic model outperforms both learned models, while it simply looks for object categories that are involved in human action history, without understanding of human goals.
- There is no further analysis on how important it is for the robot to understand human goals in task level 2 and above.
- It's unclear how good is the proposed quest generation process. Intuitively it makes sense to mimic human behavior in communications. A further analysis of data quality would be more convincing, e.g., via comparing with real human judgements.

**Additional Feedback:**

None

**Documentation:**

Yes, it has sufficient detail on benchmark creation.

**Ethics:**

No concerns

**Relation To Prior Work:**

Yes.

**Summary And Contributions:**

This paper presents a text-based robot instruction following benchmark, where the instructions have different levels of ambiguities, and the agent needs to resolve them via understanding both human intention and the physical environment.

There are existing work that also consider ambiguity in human instructions, but this work proposes a systematic way to create and control the level of ambiguity, via sampling based on the utility and cost of sub-tasks, and also based on the trade-off between precision and cost in instruction language.

Also, this benchmark is based on the tasks in BEHAVIOR-100, which has good variety in task types, object categories and actions, not a toy environment.

---

> ### Author Response · Authors · 2022-08-21
> **Response to Reviewer UhYr**
>
> **Q1**: Why does such a simple heuristic model outperform others?
>
> **A1**: The heuristic model leverages additional ground-truth information that all learning models do not have. Specifically, the heuristic model works on the symbolic state composed of object properties and relations. It also has a built-in planner to reach and manipulate objects. The only decision made by the model is to choose the proper objects to manipulate.
>
> We attribute the performance of the heuristic model because the "repeating human's action" strategy is a decent choice for using human history trajectory. By contrast, the learning models are jointly learning to represent object states, choose the target objects and solve manipulation planning problems.
>
> Another account for the good performance of the heuristic model is that, based on the environmental setup, the agent can take up to 40 moves before the episode ends. Thus, the heuristic agent can try to move multiple objects to the target location until one of them succeeded. This setting was designed to allow reinforcement learning agents to explore during training. If we constraint the model to use only one trial, as shown in the following figure, there will be a significant drop in terms of performance. That is, the heuristic agent is not a good model for resolving language ambiguities.
>
> |Heuristic model    |level1|level2|level3|level4|
> |:------------------|:----:|:----:|:----:|:----:|
> |Success Rate(%)    |94.94 |28.11 |12.02 |17.80 |
> |#Moves when Success|4.20  |4.37  |4.29  |4.36  |
>
> **Q2**: Analysis of the importance of understanding goals.
>
> **A2**: First, we would like to clarify that this paper does not suggest that understanding the goal "explicitly" is necessary for robots to complete the task. Instead, we are showing that social reasoning is an essential part of interpreting instructions, and directly inferring the goal should be one way. In the extended results (General Response Q5), we present the baseline performance when the goal or subgoal of the human is given as additional inputs. Specifically, the success rate of both fully and partially observable models (in level 2 and above) improve from around 5% to around 15% only by appending the subgoal in first-order-logic format to the inputs. This highlights that inferring human goals and subgoals is helpful in resolving ambiguous instructions.
>
> **Q3**: Human test.
>
> **A3**: Thanks for your suggestion. Per request, we have added an additional evaluation of human performance on HandMeThat.
> The results show that our human subjects can perform well on tasks in level 1 and 2, showing a success rate of more than 80% in fully-observable setting. In level 3, the inference of speaker intention gets more difficult. As a result, the performance drop significantly to 36% in fully-observable setting, but still outperforms our learning baselines. Tasks in level 4 are intrinsically under-specified. Thus, the human subjects perform worse than level 3. Success rates in partially-observable setting are also significantly lower than in fully-observable setting, suggesting that observability is also a critical challenge. Note that our environment supports robots asking additional clarification questions to the simulated human in order to resolve ambiguities. However, since no current learning models are not capable of actively asking questions, for a fair comparison, we do not enable this option in human experiments.
> Generally speaking, the human performances suggest that the generation process is reasonable, and the data splits present the hardness differences. Please refer to General Response Q4 for more details.

---

> > ### Author Response · Authors · 2022-08-26
> > **Looking Forward to Your Feedback**
> >
> > Dear Reviewer,
> >
> > Thank you again for the constructive reviews, which have helped us greatly improve the quality and clarity of our paper. In our revision, we have included several additional analyses, including the heuristics-based baseline, the importance of recognizing the goal of humans, and human performance on our task. We hope our response and new results have been able to address your concerns. As we approach the end of the discussion period, please don’t hesitate to let us know if you have any additional questions or comments!
> >
> > Thanks for your time,
> >
> > Authors

---

> > ### Comment · Reviewer_UhYr · 2022-08-28
> > **Response to authors**
> >
> > Thanks for the detailed response and the additional results. I don't have further questions and I'd like to keep the original rating.

---

### Author Response · Authors · 2022-08-21
**General Response (Part 1)**

We thank all reviewers for their thoughtful comments and suggestions. In our general response, we wish to address a few questions that were raised by multiple reviewers.

**Q1**: The task setting simplifies the real HRI cases.

**A1**: We agree with reviewers that there are still gaps between our benchmark and real-world scenarios, such as visual perception challenges and non-verbal communications. We have included them in our discussion. However, we would like to clarify that HandMeThat is not designed for exactly model real-world human-robot interaction. In other words, we do not expect training models solely on HandMeThat and directly apply them in real-world scenarios. Just like many benchmarks in embodied AI [1, 2], HandMeThat should be used as a diagnostic benchmark for developing models and inspecting different aspects of models. Thus we have been focusing on a systematical evaluation of models that can interpret ambiguous language instructions considering both the physical contexts and actions and goals of humans. For example, because all trajectories and language were generated with programs, we naturally have annotations at various levels (human's internal goals, the grounded meaning of instructions, etc.) This enables us to split the dataset into different hardness levels and also compare models with different supervisions (see our General Response Q5 for details).

[1] Srivastava et al. BEHAVIOR: Benchmark for Everyday Household Activities in Virtual, Interactive, and Ecological Environments. In CoRL, 2021.

[2] Shridhar et al. ALFWorld: Aligning Text and Embodied Environments for Interactive Learning. In ICLR, 2021.

**Q2**: The textual environment seems quite different from the visual domain.

**A2**: We agree with reviewers that the textual interface, to some degree, simplifies the perception and recognition problems. However, HandMeThat still illustrates the key challenges in understanding ambiguous language instructions. In fact, rendering in textual interface bypasses the perception and recognition difficulties, and allows us to focus on the key challenges in instruction reasoning.

We can break down the challenges of "HandMeThat" into four subtasks: 1) perceives and represents the world state, 2) pragmatically reasons the instruction, 3) makes a plan and chooses the proper next action, and 4) interacts with the world. The differences between textual and visual interfaces would affect the design of the first and last subtask, but our benchmark aims at evaluating the ability of the middle two subtasks. The current textual interface we use is alleviating the difficulty of perceiving and representation. It's even possible to feed all the information to the model in a symbolic way since such challenges are not what we are studying.

Furthermore, it's theoretically possible to render the HandMeThat tasks into the visual domain, but we haven't moved to that due to some technical issues. Several challenges in applying the iGibson2.0 or AI2-THOR simulator include 1) defining the goal, 2) rendering enormous object space with various attributes, and 3) representing human trajectory. If the task can be properly rendered into the visual domain in the future, we can append other challenges in perceiving and object manipulation to the pipeline.

**Q3**: What's the need of this benchmark? How will the (HRI) community benefit from it?

**A3**: While there has been extended literature on pure planning tasks as well as textual-based RL tasks, the existing benchmarks cannot evaluate the abilities we are focusing on---resolving ambiguous instructions based on both the physical world state and actions and goals of another agent (human). Developing a benchmark for such scenarios is challenging in both data collection and automatic evaluation. Specifically, it's hard to collect the everyday dialogues with corresponding physical states; and it is hard to build automatic evaluation protocols that involve human judgments about robot success. HandMeThat is a first step toward building such a benchmark: it leverages the availability of human-annotated real-world task distributions (BEHAVIOR) and computational models for human language modeling (RSA models) and creates a benchmark for contextual language understanding with physical and social information (i.e., actions and goals of humans) that supports automatic data generation and evaluation.

Another important feature of the proposed benchmark is that it supports fine-grained "diagnostics" of models. HandMeThat disentangles different challenges by splitting the dataset into four hardness levels. The challenges include planning, goal inference, and pragmatic language reasoning. The data split here can be separately used for studying different modules.

Finally, since the episodes in our benchmark can be symbolically represented, it can be potentially rendered into different interfaces, such as images, too.

---

### Author Response · Authors · 2022-08-21
**General Response (Part 2)**

**Q4**: Human performance on the tasks.

**A4**: We collect human performance on each hardness level. We first introduce human subjects to the benchmark settings including possible locations, objects, and tasks. Next, human subjects interact with our textual interface. For both fully- and partially-observable settings, we collect 25 episodes of human performance for each hardness level. Human subjects are only allowed to choose one particular object to manipulate and stop immediately after execution no matter if the goal is reached.
The results are shown as followings:

|Fully Observable    |level1|level2|level3|level4|
|:-------------------|:----:|:----:|:----:|:----:|
|Success Rate(%)     |92    |80    |36    |16    |
|#Moves when Success |4.4   |4.8   |4.3   |4.5   |

|Partially Observable|level1|level2|level3|level4|
|:-------------------|:----:|:----:|:----:|:----:|
|Success Rate(%)     |76    |52    |20    |8     |
|#Moves when Success |5.1   |4.8   |5.8   |5.0   |

The results show that our human subjects can perform well on tasks in level 1 and 2. In level 3, the inference of speaker intention gets more difficult. As a result, the performance drops significantly, but still outperforms our learning baselines. Tasks in level 4 are intrinsically under-specified. Thus, the human subjects perform worse than level 3. Note that our environment supports robots asking additional clarification questions to the simulated human in order to resolve ambiguities. However, since no current learning models are not capable of actively asking questions, for a fair comparison, we do not enable this option in human experiments.
Generally speaking, most of the HandMeThat tasks in first three levels can be solved without acquiring more knowledge. Thus, the relatively poor performance of our learning baselines suggests significant room for improvement.

**Q5**: Extended experiment results.

**A5**: As suggested by some reviewers, we have conducted more experiments. The following tables show the success rate of updated baselines. More details can be found in the Experiments section of our revised paper.

Fully observable:
|Success Rate|level1|level2|level3|level4|
|:-----------|:----:|:----:|:----:|:----:|
|Seq2Seq     |22.44 |6.83  |4.01  |5.51  |
|+ goal      |17.90 |10.44 |6.98  |8.05  |
|+ subgoal   |25.88 |13.25 |13.95 |15.25 |

Partially observable:
|Success Rate|level1|level2|level3|level4|
|:-----------|:----:|:----:|:----:|:----:|
|Seq2Seq     |25.49 |10.44 |3.88  |5.30  |
|+ goal      |21.01 |8.24  |7.36  |5.51  |
|+ subgoal   |21.99 |15.66 |11.44 |13.35 |

- We optimize the architecture of Seq2Seq model. For fully-observable models, we implement a separate encoder to encode the full observation strings besides another encoder for partial information. We train a partially-observable model on this updated architecture, where both encoders will only receive the partial observation strings. We also tune the models and train for a longer time. As a result, the performances improve significantly for both the fully- and partially-observable setting.
- Interestingly, for level 1 and 2, the Seq2Seq model shows slightly better performance in the partially-observable setting, although the expert demonstration is the same. We attribute this to the network capacity issue. Since we kept the model size to be the same, the model for the partially-observable setting has doubled the parameters for encoding task-relevant information such as the objects at the current location.
- As a reference, we also provide the performance of Seq2Seq model given extra oracle information (goal or subgoal of the human agent in first-order-logic format). The formula is directly provided along with the utterance in input texts. The results show that all the performances except level 1 significantly improve. This highlights that the inference of human goals and subgoals is helpful in resolving ambiguous instructions.
- We apply the reward shaping method in training DRRN models. However, training with this modified reward function does not improve overall performance. That is, the overall success rate is still 0.00\% for all the settings. When we visualize the predicted actions of the agent, we find that in many cases the models manage to predict the first or second action correctly, but they still fail to complete the whole task. We believe that integrating imitation learning and reinforcement learning such as DAgger, is needed to tackle HandMeThat tasks.

---

### Meta-Review · Area_Chair_79kn · 2022-09-09

**Recommendation:** Accept
**Confidence:** 3

**Metareview:**

The authors presented a new text-based benchmark environment where agents are challenged to follow ambiguous instructions to help humans achieve their goal.

### Review Summary

#### Strengths:

- Almost all reviewers agreed that the paper provided an important novel contribution by focusing on a benchmark for collaborative agents working under ambiguity.
- Several reviewers praised the solid construction of the benchmark, in particular the utterances and sub-goal generation.
- Some reviewers also praised the clarity of the paper, particularly the description of the generation process, noting it would be useful to other researchers. This clarity was further improved by significant revisions, including a more formal problem statement and description and a discussion of the different challenges posed in this task.

#### Weaknesses:

- Several reviewers criticized the experimental section, often specifically requesting additional details of human performance, and better tuning.  These were added by the authors in a subsequent revision, along with revised results for Seq2Seq model with improved tuning, and discussion of the heuristic model performance.
- Several reviewers also criticized the environment for being too simple, being a text-based environment.  This was met with a convincing argument about the value of separating visual understanding from pragmatic reasoning and planning. This ‘simplicity’ argument was also undermined by the large performance gap learned methods had compared to human performance.
- One reviewer engaged very thoroughly with the paper, criticising its lack of formalism, interrogating the problem statement and questioning its contribution to the wider academic landscape. The discussion that ensued proved very illuminating, with substantial revisions and discussions that addressed many of these issues, though did not satisfy the reviewer enough to recommend acceptance.

### AC View

Following the reviews, discussions, and substantial improvements made to this paper I have decided to accept this paper, on account of the generally accepted importance and novelty of this work and the solid construction and clear presentation of it.  The revisions made to the paper during the discussion period made a _significant contribution_ to this decision, not least in greatly improving the clarity of the work and helping place it in the wider academic context. Indeed, this process prompted several other reviewers to raise their scores.

This view reflects the majority of reviewers. While one reviewer recommended a narrow rejection, after a very thorough and very useful review, I did not find their remaining criticisms sufficient to reject the paper.

---

### Decision · Program_Chairs · 2022-09-16

Accept